Citation: *Molecular Systems Biology* 9:651
www.molecularsystemsbiology.com

# Temporal system-level organization of the switch from glycolytic to gluconeogenic operation in yeast

Guillermo G Zampar[1], Anne Kümmel[2], Jennifer Ewald[2], Stefan Jol[2], Bastian Niebel[1], Paola Picotti[2], Ruedi Aebersold[2,3], Uwe Sauer[2], Nicola Zamboni[2] and Matthias Heinemann[1,2,*]

[1] Molecular Systems Biology, Groningen Biomolecular Sciences and Biotechnology Institute, University of Groningen, Groningen, The Netherlands, [2] ETH Zurich, Institute of Molecular Systems Biology, Zurich, Switzerland and [3] Faculty of Science, University of Zurich, Zurich, Switzerland
* Corresponding author. Molecular Systems Biology, Groningen Biomolecular Sciences and Biotechnology Institute, University of Groningen, Nijenborgh 4, 9747 AG Groningen, The Netherlands. Tel.: +31 50 363 8146; Fax: +31 50 363 4165; E-mail: m.heinemann@rug.nl

The diauxic shift in *Saccharomyces cerevisiae* is an ideal model to study how eukaryotic cells readjust their metabolism from glycolytic to gluconeogenic operation. In this work, we generated time-resolved physiological data, quantitative metabolome (69 intracellular metabolites) and proteome (72 enzymes) profiles. We found that the diauxic shift is accomplished by three key events that are temporally organized: (i) a reduction in the glycolytic flux and the production of storage compounds before glucose depletion, mediated by downregulation of phosphofructokinase and pyruvate kinase reactions; (ii) upon glucose exhaustion, the reversion of carbon flow through glycolysis and onset of the glyoxylate cycle operation triggered by an increased expression of the enzymes that catalyze the malate synthase and cytosolic citrate synthase reactions; and (iii) in the later stages of the adaptation, the shutting down of the pentose phosphate pathway with a change in NADPH regeneration. Moreover, we identified the transcription factors associated with the observed changes in protein abundances. Taken together, our results represent an important contribution toward a systems-level understanding of how this adaptation is realized.
*Molecular Systems Biology* **9**: 651; published online 2 April 2013; doi:10.1038/msb.2013.11
*Subject Categories:* metabolic and regulatory networks; cellular metabolism
*Keywords:* diauxic shift; fluxome; metabolome; proteome; *Saccharomyces cerevisiae*

## Introduction

In response to nutrient availability, cells rearrange their metabolism to ensure survival. One of the most central metabolic rearrangements is the transition from a glycolytic to a gluconeogenic carbon source that involves redirection of the carbon flux through the Embden-Meyerhoff pathway. Despite the wide-spread relevance of this glycolysis-to-gluconeogenesis adaptation from microbes to mammalian cells (Wasserman and Cherrington, 1991; Chandramouli *et al*, 1997), we lack a system-level understanding about the regulatory mechanisms that govern this transition.

A particularly good model system for studying this nutritional adaptation in eukaryotic cells is the budding yeast *Saccharomyces cerevisiae* (Botstein and Fink, 2011). When cultivated in high glucose conditions, *S. cerevisiae* operates in glycolytic mode to largely ferment the available sugar to ethanol, independent of the presence of oxygen (Dickinson and Schweizer, 1999). Once glucose is depleted, the cells consume the earlier produced ethanol by switching to gluconeogenesis and concomitantly increasing their respiration rate, which is generally believed to be a consequence of a tricarboxylic acid (TCA) upregulation (Brauer *et al*, 2005).

This change from growth on glucose to growth on ethanol is known as the 'diauxic shift' (Dickinson and Schweizer, 1999).

The major source of information about the regulation of the diauxic shift in yeast comes from transcriptome studies. During the shift, the abundances of over 1700 transcripts change involving distinct changes already before and after glucose depletion (DeRisi *et al*, 1997; Hanisch *et al*, 2002). A model was proposed in which cells undergo progressive changes before glucose depletion and abruptly remodel their metabolism upon glucose exhaustion. This abrupt reorganization is followed by a period of progressive adaptation to growth on ethanol (Brauer *et al*, 2005). Thus, as suggested by Radonjic *et al* (2005), the diauxic shift is a complex process that requires metabolic changes before, upon and after the exhaustion of glucose. Comparative proteome analyses using two-dimensional gel electrophoresis suggested the involvement of several transcription factors including Msn2p and Msn4p (Boy-Marcotte *et al*, 1998), Cat8p (Haurie *et al*, 2001) and Sip4p (Vincent and Carlson, 1998). The known upstream events that trigger these changes include a drop in cAMP levels (Boy-Marcotte *et al*, 1996; Garreau *et al*, 2000) and protein kinase A (PKA) activity (Enjalbert *et al*, 2004; Roosen *et al*, 2005), the activation of the Snf1 (Enjalbert *et al*, 2004; Haurie

*et al*, 2004) pathway and the inactivation of the target of rapamycin (TOR) pathway (Slattery *et al*, 2008).

While several regulation mechanisms involved in the diauxic shift are known, we do not known how and when they control the physiological adjustment of metabolic fluxes. In fact, we do not even know the exact dynamic changes of intracellular fluxes during the diauxic shift. In this work, we generated a unique set of dynamic and quantitative omics data during the diauxic shift including an extensive characterization of the extracellular metabolite concentrations, and intracellular metabolome and proteome data. From these dynamic data sets, we identified the three main events that lead to the adaptation and pinpointed causal molecular regulations that drive the observed changes in metabolic fluxes. In addition to contributing to our understanding of the extensive remodeling of metabolic fluxes in this particular case, our study is also an example of how the integration of large-scale experimental data can generate understanding about complex biological processes.

## Results and discussion

### Temporal organization of the diauxic shift

To unravel the physiological changes that cells undergo during the diauxic shift, we first captured the dynamics of the abundance of extracellular metabolites (glucose, ethanol, pyruvate, succinate, acetate and glycerol) and the biomass concentration ($OD_{600}$). With these data, we estimated the time courses of the specific uptake and excretion rates (Figure 1A–H, cf. Supplementary File 1 for full data set). Throughout the adaption, which we found to span over 7 h (Figure 1A–H), we identified different 'phases' with specific physiological states.

The first change already occurs 1.5 h before glucose depletion with a 20-fold drop in the specific $CO_2$ production rate (Figure 1H, blue curve) and a slight concomitant decrease in the specific succinate (Figure 1D, blue curve), ethanol (Figure 1E, blue curve), and glycerol (Figure 1F, blue curve) excretion rates (28, 26 and 23%, respectively, between − 2.1 and − 0.6 h). On the contrary, our data does not indicate a reduction in the rate of biomass production and glucose uptake rate (Figure 1A and C, blue curves). After glucose exhaustion, there is a sudden decrease in the rate of biomass production (Figure 1A, blue curve) and cells start consuming ethanol (Figure 1E). Between 0 and 2 h (after glucose exhaustion), pyruvate consumption is very high (0.3 mmol/h gDW) until its depletion (Figure 1G, blue curve), after which acetate production rate decreases and stops completely at 6 h (Figure 1B, blue curve). Thereafter, cells enter a new steady state characterized by constant growth on ethanol. These physiological changes are consistent with the previously developed model (Brauer *et al*, 2005) in the sense that there are gradual adaptations already before glucose depletion and that an abrupt remodeling occurs when glucose is exhausted, followed by an additional stage of continuous metabolic adaptation afterwards.

To assess the intracellular counterpart of these observed physiological changes, we measured the concentrations of 69 metabolites from central and amino-acid metabolism as well as the concentrations of the energy and redox cofactors (Figure 2A–C; Supplementary File 2). These measurements revealed a characteristic behavior of intracellular metabolite concentrations according to the pathways in which they are involved. While the levels of metabolites in the glycolysis/gluconeogenesis and pentose phosphate (PP) pathway drop after glucose depletion (Figure 2A and C), the concentrations of TCA/glyoxylate cycle metabolites increase significantly (Figure 2B). The only exception to these general patterns is phosphoenolpyruvate (pep) (cf. inset in Figure 2A). The accumulation of phosphoenolpyruvate is related to the mechanism, by which cells revert the metabolic fluxes through glycolysis (see below, 'Preparing the reversion of metabolic fluxes').

Next, we asked to what extent the metabolic fluxes are readjusted and at which time point. Since experimental quantification of intracellular fluxes is neither possible for dynamic systems nor for growth on multiple carbon sources, we used flux variability analysis (Mahadevan and Schilling, 2003) to estimate the possible flux ranges at the different time points using our experimental data as constraints. Specifically, the computational optimization problem, drawing on a 240-reaction stoichiometric model of yeast central metabolism adapted from Jol *et al* (2012) (cf. Supplementary File 4), was constrained by the determined uptake and excretion rates (Figure 1, blue curves) and by the determined concentrations of intracellular metabolites (Figure 2A–C; Supplementary File 2), which were coupled to the model via the second law of thermodynamics (Kümmel *et al*, 2006; Henry *et al*, 2007; Hoppe *et al*, 2007). We further constrained the solution space by setting a lower limit to ATP turnover on the basis of the idea that evolution has optimized for energetic efficiency (Schuetz *et al*, 2007) (cf. Materials and methods).

To identify the flux ranges in the different stages of the adaptation, we performed the flux variability analysis at five different characteristic time points. Even though, strictly speaking, the steady-state assumption required for flux variability analysis holds only for the first and last time points, we found that the concentrations of intracellular metabolites are nearly constant at the intermediate time points as well (specially the glycolytic and PP pathway intermediates, and to a lesser extent the TCA/glyoxylate cycle metabolites, Figure 2A–C and Supplementary File 2), which indicates that flux variability analysis is a good approximation at those time points also. As the composition of cells can vary during the diauxic shift (e.g., in terms of the amount of storage compounds; Francois and Parrou, 2001), we included exchange reactions in our model for the production/consumption of storage carbohydrates thereby adding an additional degree of freedom to the model to account for dynamically varying contents of storage carbohydrates (Supplementary File 3.1).

Figure 3A–E shows the determined intracellular flux ranges at the different time points, with the thickness of the arrows being proportional to the average of the maximal and minimal possible flux values through each reaction. Figure 3F shows the exact flux ranges for a set of reactions in central carbon metabolism; the flux ranges for the remaining reactions can be found in Supplementary File 5. The dynamic flux patterns reveal that the reorganization of metabolic operation

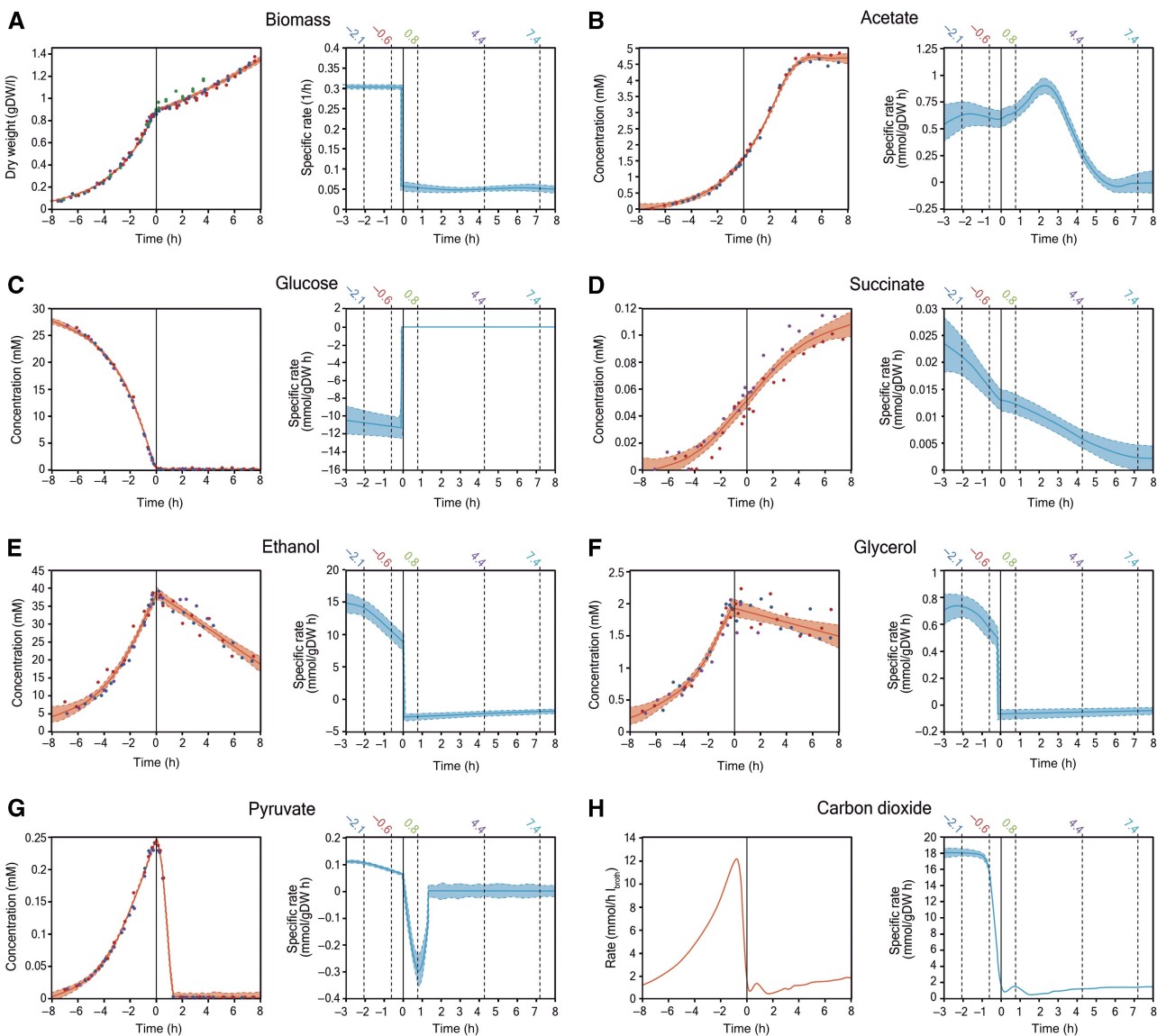

**Figure 1** (**A–H**) Extracellular metabolite levels and uptake/excretion rates. The experimental measurements (dots) were fitted (orange curves) as described. Different colors in the scatter plots represent individual biological replicates. In the case of carbon dioxide production rate, the individual measurements are not shown because there are over 10 000 data points. The regressions curves were derived and divided by the cellular dry weight at each time point to calculate the specific rate of biomass production and the specific rates of uptake/excretion of each metabolite (shown in blue). Shaded areas represent the 95% confidence intervals for each curve and the vertical dashed lines show the time points chosen for the determination of intracellular flux ranges and Gibbs-free energies of reaction. Glucose depletion at $t = 0$ h is indicated with a vertical solid line.

is accomplished by three major events that take place progressively throughout the adaptation process. These are (i) a redistribution of fluxes toward storage compounds and a decrease in the glycolytic flux before glucose depletion, (ii) the reversion of fluxes through glycolysis after the exhaustion of glucose with an increase in glyoxylate cycle activity, and (iii) a shift in NADPH regeneration during the last hours of the adaptation process (Figure 3, I–III). In the following, we explore the mechanisms that accomplish these changes.

## Preparing the reversion of metabolic fluxes

The first event in the metabolic adaptation, even occurring before glucose depletion, consists of two intertwined processes: the reduction in flux through glycolysis (22% decrease between $-2.1$ and $-0.6$ h) and redirection of the taken-up glucose toward storage compounds (600% increase between $-2.1$ and $-0.6$ h) (Figure 3A–B, indicated by 'I'). To obtain a first idea of how the reduction in the glycolytic flux is accomplished, we estimated the Gibbs-free reaction energies for each metabolic reaction at each time point using our generated metabolome data (Supplementary File 5). Gibbs-free reaction energies were suggested to harbor information about active regulation occurring at single enzymatic reactions; that is, reactions that operate far from equilibrium are more likely to be actively regulated (Crabtree and Newsholme, 1978; Kümmel *et al*, 2006). The identified key regulatory spots are depicted in Figure 3A–E with filled boxes.

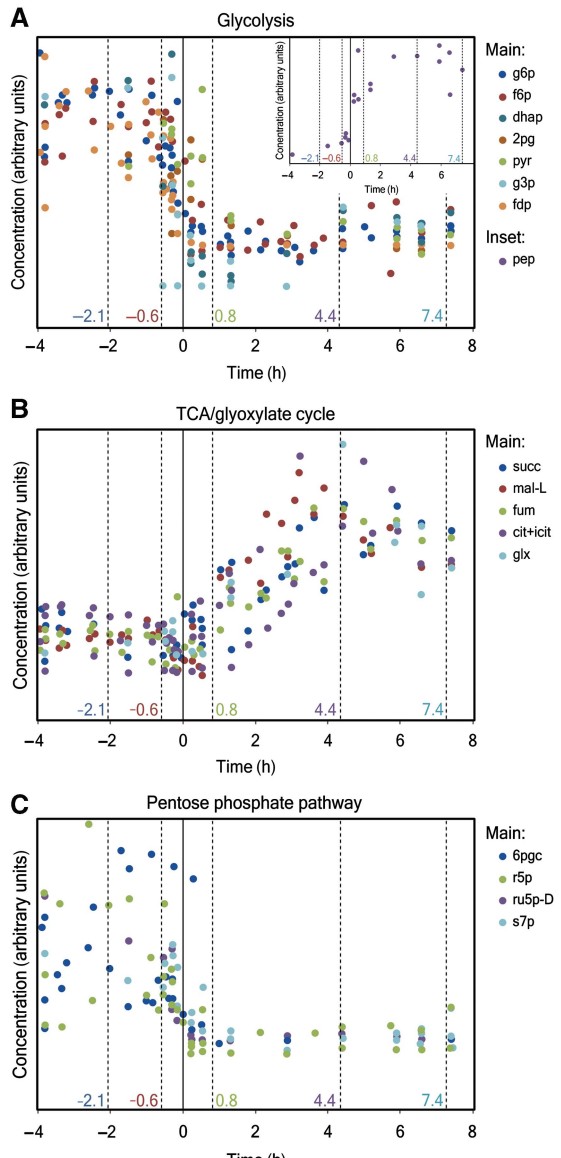

**Figure 2** (**A–C**) Intracellular concentrations of metabolites participating in glycolysis (A), the TCA/glyoxylate cycle (B) and PP pathway (C). The concentrations of each metabolite were scaled appropriately to show the general trend in each pathway. Phosphoenolpyruvate, which presents a different behavior than the general trend, is shown in the inset of (A). The time points chosen for the determination of intracellular flux ranges and Gibbs energies of reactions are depicted with dashed lines. Glucose depletion ($t = 0$ h) is indicated with solid vertical lines. Further information (abbreviations used and the actual values of concentrations for each metabolite) is available in Supplementary File 2.

Through this analysis, we found that the phosphofructokinase (PFK) and pyruvate kinase (PYK) reactions move farther away from equilibrium between $-2.1$ and $-0.6$ h (Figure 3B), indicating a decrease in these reactions' catalytic activities, which suggests active repression or inactivation. The repression/inactivation of these reactions most likely has a dam-like effect restricting the flux through the glycolytic pathway and resulting, for instance, also in the accumulation of the PYK's substrate, phosphoenolpyruvate (Figure 2A, inset). The reduced glycolytic flux causes the concentrations of all other glycolysis metabolites to decrease (Figure 2A, main)

and the Gibbs-free energies of the rest of glycolytic reactions—enolase (ENO), glyceraldehyde-3-phosphate dehydrogenase (GAPD), glucose-6-phosphate isomerase (PGI) and phosphoglycerate kinase (PGK) reactions—move closer to zero (Supplementary File 5). Thus, the flux reduction through glycolysis before glucose is depleted moves these reactions closer to equilibrium and thereby prevents a later abrupt transition, providing a mean to prepare for the upcoming necessary flux reversal.

The inferred reduced catalytic activity of the PFK and PYK reactions must be realized either by regulation on the enzymatic level or by changes in enzymes' expression levels. To assess the relevance of enzyme abundance changes during the adaptation process, we measured the abundance of 72 enzymes of central metabolism at characteristic time points during the shift (Supplementary File 6). According to these data, there is a slight but significant drop in 6-phosphofructo-kinase (Pfk1p, the PFK-catalyzing enzyme) levels (fold change between $-0.1$ and $-1.7$ h = 0.62, $P$-value = 0.003), which could account for the observed shift in this reaction's Gibbs-free energy. In turn, while the expression of other glycolytic enzymes is also downregulated before glucose depletion (Figure 4A, 'Glycolysis'), the observed shift in Gibbs-free energy of the PYK reaction seems not to be caused by downregulation of protein expression as the abundance of pyruvate kinase 1 (Cdc19p, the PYK-catalyzing enzyme) is basically unaltered (Supplementary File 6) suggesting that the regulation of PYK must rather be of allosteric nature or due to post-translational modifications such as phosphorylation (Oliveira and Sauer, 2012). In fact, it has been shown that PKA activity has a direct effect on the activity of Cdc19p (Portela *et al*, 2002); thus, the decrease in PKA activity triggered by the drop in glucose concentration (Tamaki, 2007) may restrict the flux through glycolysis at the PYK level.

Simultaneous to the decrease in glycolytic flux (and probably as a consequence of this reduction), part of the taken-up glucose is redirected toward the accumulation of the storage carbohydrates trehalose and glycogen, as revealed by our flux analysis (cf. Supplementary File 3.2). This redistribution is evidenced by the non-zero flux through the reaction that leads to the production of reserve compounds at $-0.6$ h (cf. Figure 3A, B and F, 'Storage'). The higher production of storage carbohydrates is consistent with the previously described increase in glycogen and trehalose content of yeast cells during the diauxic shift (Francois and Parrou, 2001). Also, the flux redistribution is consistent with the changes in protein expression patterns that we observed between exponential growth and right before glucose exhaustion: the cluster of enzymes associated with the metabolism of reserve carbohydrates peaks before glucose exhaustion and then returns slowly toward basal levels over the rest of the adaptation (Figure 4A, indicated by 'I' and 'Starch metabolism'). For example, the phosphoglucomutase 2 enzyme (Pgm2p), which catalyzes the conversion of glucose-6-phosphate to glucose-1-phosphate, is maximal right before glucose depletion (Supplementary File 6).

Interestingly, the flux variability analysis does not reveal a net consumption of reserve carbohydrates after glucose exhaustion (as evidenced by the positive flux ranges through the 'Storage' reaction in Figure 3F), which strongly suggests

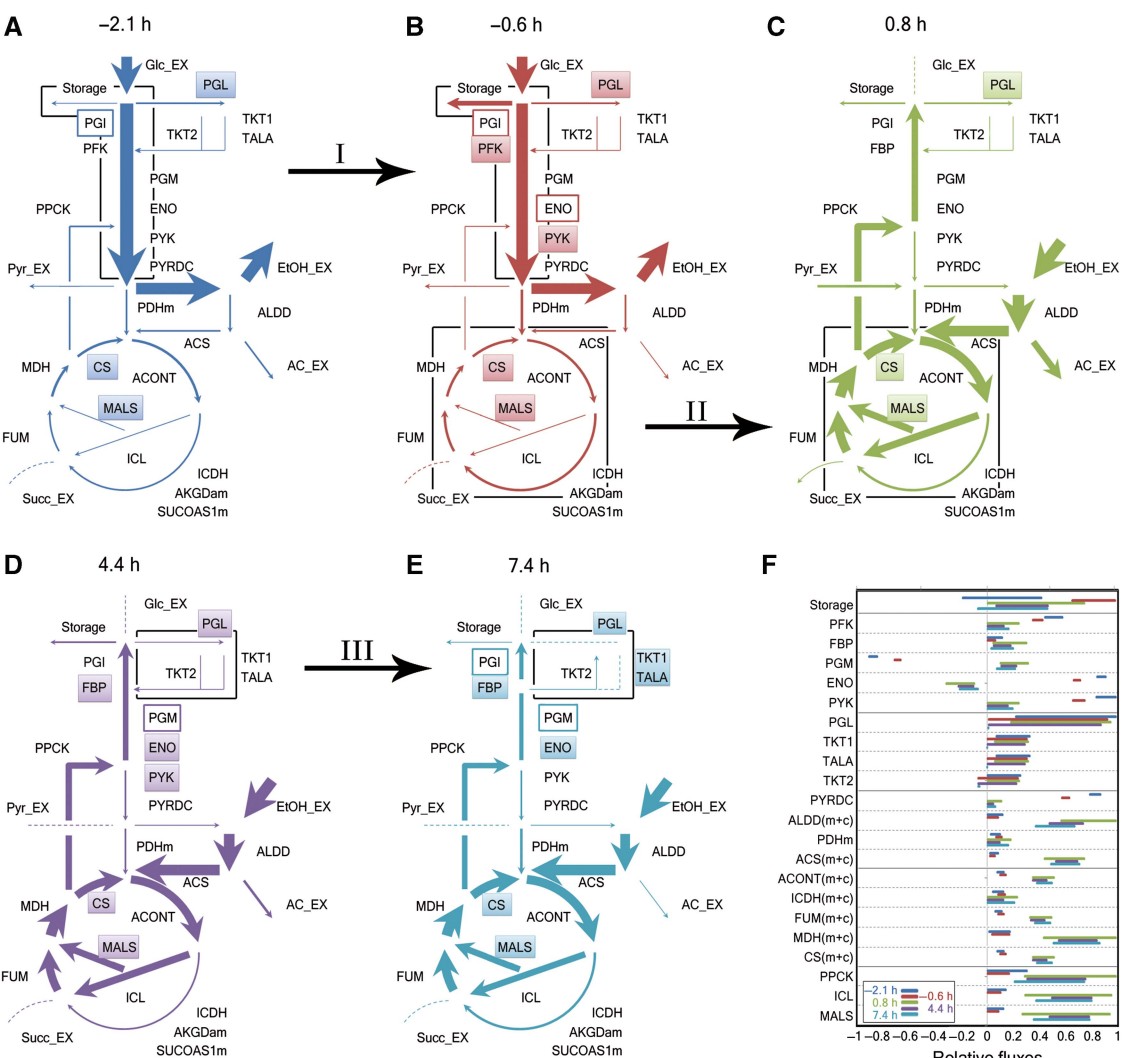

**Figure 3** (**A–E**) Graphical representation of the metabolic flux distribution in the central carbon metabolism at the indicated time points. Reaction names are taken from the model used for the flux variability analysis (Supplementary File 4). The arrow thickness is related to the average of the maximal and minimal value of the flux through each reaction relative to the growth rate at each time point, as follows: dashed lines for fluxes lower than 0.2 mmol/gDW, for values between 0.2 and 20 mmol/gDW the thickness is proportional to the flux and for fluxes higher than 20 mmol/gDW the thickness increments every 5 mmol/gDW (exact values are provided in Supplementary File 5). Reactions that are far from equilibrium (i.e., with a Gibbs-free energy of at least 5 kJ/mol in either direction) are indicated with filled boxes, while empty boxes denote reactions that are close to equilibrium (i.e., with a Gibbs-free energy of at most 5 kJ/mol in both directions). The three main events of the adaptation process are indicated with roman numerals: I, decreased glycolytic flux and production of storage compounds; II, onset of the glyoxylate cycle; III, halting of the PP pathway. (**F**) Flux ranges through the reactions shown in (A–E). The reactions are grouped according to the pathway to which they belong and each group is separated by solid lines. The pathways, from top to bottom, are alternative carbon metabolism, glycolysis/gluconeogenesis, PP pathway, pyruvate metabolism, TCA/glyoxylate cycle and anaplerosis. The relative fluxes were scaled within each group to fit the interval between − 1 and 1, taking into account the maximal and minimal values within each pathway. The maximal and minimal values for the reactions labeled as '(m + c)' make reference to the net flux through that reaction in the cytoplasm plus the same reaction in the mitochondria, and the fluxes labeled as 'Storage' represent the amount of glucose residues being consumed (or produced) by trehalose and glycogen synthesis (or hydrolysis).

that the accumulated glycogen is not used as a carbon source when glucose is depleted as it is usually mentioned (Francois and Parrou, 2001). Instead, the redirection of fluxes toward storage compounds is most likely a consequence of the reduction in the flux through glycolysis right before glucose depletion (Supplementary File 3.2).

Next, we aimed at identifying the transcription factors that are responsible for accomplishing the observed protein expression changes. Therefore, we performed an over-representation analysis considering the protein abundance changes between each two consecutive time points and using data on transcription factor-gene interactions (Teixeira *et al*, 2006). This analysis identifies transcription factors associated with the measured proteins that are over-represented in the subset of regulated proteins. It is important to note that the over-representation analysis does not provide a complete list of all transcription factors involved, but rather identifies those transcription factors, which realize the changes in abundances of the measured proteins.

Here, we found that the changes before glucose depletion are likely accomplished by the activation of the transcription factors Msn4p, Pdr3p, Ino2p and Hsf1p (Figure 4D;

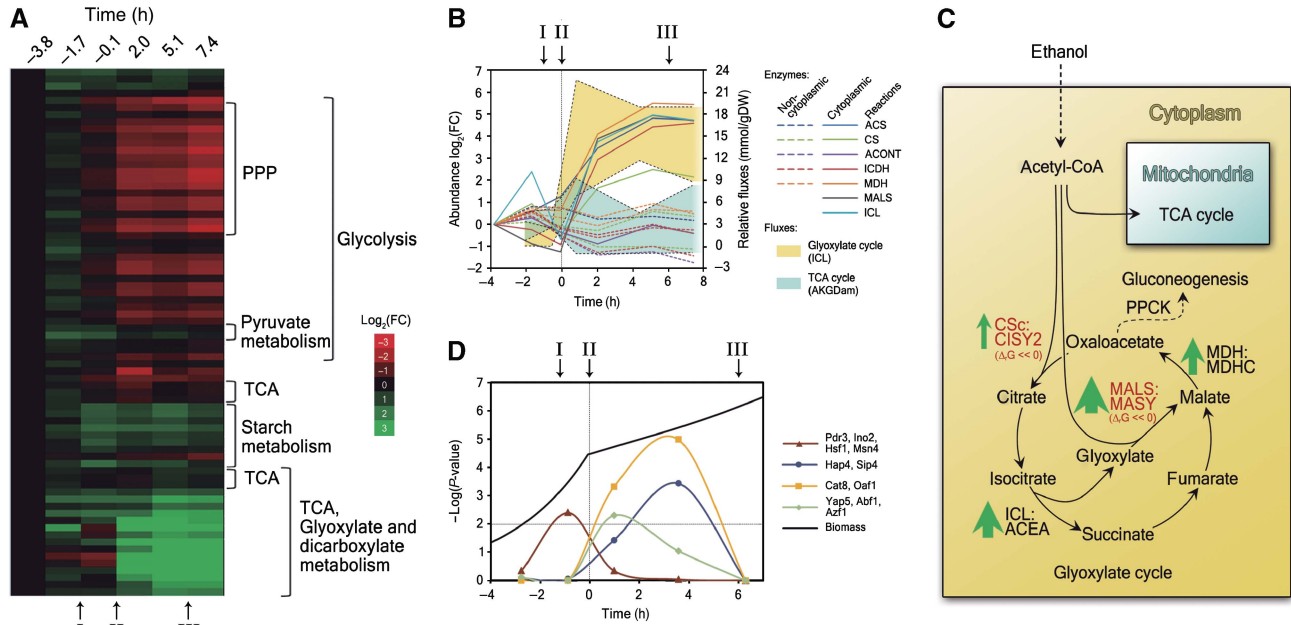

**Figure 4** (**A**) Heat map showing the changes in protein abundance at different time points. Colors show the degree of upregulation (green) or downregulation (red) in terms of the logarithm to base two of the fold change with respect to the reference state ($t = -3.8$ h). Brackets indicate clusters that are enriched in enzymes from particular pathways (Supplementary File 6). (**B**) Comparison between the expression levels of TCA/glyoxylate cycle enzymes that are localized in the cytoplasm (solid lines) and those that are not (dashed lines). Colors indicate the reactions catalyzed by those enzymes. The shaded areas represent the flux ranges through ICL (a glyoxylate cycle-specific reaction) in yellow and through AKGDam (a TCA cycle-specific reaction) in cyan at different time points. (**C**) Schematic representation of the reactions in the glyoxylate cycle and its regulation. Reactions in red were found to be far from equilibrium at all time points. The thickness of the green arrows indicates fold change of protein abundances upon glucose exhaustion. (**D**) $-\log(P\text{-value})$ of the most significant transcription factors as revealed by the hypergeometric mean test (over-representation). Relevant transcriptions factors were grouped according to their behavior during the diauxic shift (see Supplementary File 7) and the average $-\log(P\text{-value})$ of each group was plotted against time. The horizontal dotted line denotes the threshold corresponding to a $P$-value of 0.01. Glucose exhaustion and the biomass curve are also shown. Roman numerals in (A, B and D) indicate the occurrence of the three main events during the adaptation (see text).

Supplementary File 7). These transcription factors are associated with a wide range of stress responses, including osmotic stress and acidic stress (Wu and Chen, 2009) and their main target genes are related to storage compound accumulation, upregulation of Pgm2p, hexokinase 1 (Hxk1p) and glucokinase 1 (Glk1p). While Msn4p was already earlier suggested to be involved in the diauxic shift (Boy-Marcotte *et al*, 1998), Pdr3p, Ino2p and Hsf1p were so far not associated with this process. In fact, these three transcription factors are linked with 80% of the proteins that are upregulated just before glucose depletion (Supplementary File 7). To further support our predictions with experimental data, we constructed the deletion mutants of Msn4, Pdr3 and Ino2 (the deletion mutant of Hsf1 is not viable). Indeed, the ΔMsn4 and ΔPdr3 strains show a different diauxic shift behavior than the wild type, while ΔIno2 also had a growth defect on glucose (Supplementary Files 3.4 and 7). Even with these indications, ultimately proving the involvement of these transcription factors requires further experimental testing.

### Fueling gluconeogenesis

After glucose is depleted, yeast uses the ethanol that was secreted earlier. In the first 90 min after glucose exhaustion, pyruvate is also significantly consumed, while acetate excretion is observed till later time points (for a discussion on pyruvate and acetate utilization, see Supplementary File 3.3). Generally, it is believed that, under ethanol

conditions, an increase in the metabolic flux through the TCA cycle would provide the necessary NADH to fuel respiration (Brauer *et al*, 2005), while the glyoxylate cycle would provide the precursors for gluconeogenesis. As most of the reactions in the TCA cycle are shared with the glyoxylate cycle, we could not resolve the absolute fluxes through these two pathways individually but we could only estimate the flux ranges through the lumped reactions of the TCA (present in the mitochondria) and glyoxylate cycle (localized in the cytoplasm; Duntze *et al*, 1969) (reactions denoted as 'm + c' in Figure 3F). To assess the relevance of the TCA and the glyoxylate cycle after glucose depletion, we analyzed the abundances of pathway-specific enzymes.

Examination of the protein expression data after the carbon source shift revealed that—even though there was a large 'TCA cycle' cluster whose abundance increases drastically upon glucose exhaustion—two smaller subclusters remained constant or even dropped (Figure 4A, 'TCA'). In fact, a differential behavior of isoenzymes that catalyze reactions in the TCA/glyoxylate cycle became obvious. We found that this differential expression of isoenzymes was related to their subcellular localization: the cytoplasmic variants were upregulated, while the abundance of the non-cytoplasmic (i.e., mitochondrial) variants remained constant or even dropped (Figure 4B). For example, while the cytoplasmic isoenzyme of the citrate synthase reaction (CS), Cit2p, increased from 2.0 h on, Cit1p (mitochondrial) remained constant and Cit3p (mitochondrial) dropped (Supplementary

File 6). Similar differences were observed for other proteins associated with the TCA/glyoxylate cycle, that is, the ones that catalyze the acetyl-CoA synthetase (ACS), aconitase (ACONT), isocitrate dehydrogenase (ICDH) and malate dehydrogenase (MDH) reactions (Figure 4B). Localization data were taken from the Yeast GFP Fusion Localization Database (Huh *et al*, 2003).

These increases in cytoplasmic proteins after glucose depletion suggest increased fluxes through the glyoxylate cycle. Even though we could not identify the absolute fluxes of the reactions that are shared between the TCA and glyoxylate cycles, the reactions that are specific to one of these two pathways provide a good indication of the overall flux through each cycle. In fact, the estimated relative flux through the oxoglutarate dehydrogenase reaction, AKGDam (a reaction only involved in the TCA cycle), did not change significantly among the different time points (estimated ranges changed from 2.6–5.1 to − 1.0–9.2 mmol/gDW before and after glucose depletion, respectively), while the relative flux through the cytoplasmic isocitrate lyase reaction, ICL (a glyoxylate cycle-specific reaction), presented a steep increase upon glucose exhaustion (ranges change from 0.04–2.5 to 6.8–22.6 mmol/gDW) (Figure 4B). Importantly, the glyoxylate cycle does not only have the function of fueling gluconeogenesis, but it also has a significant role in NADH production, surpassing even the contribution of the TCA cycle after glucose depletion. Specifically, after the exhaustion of glucose the maximal TCA cycle contribution toward the total NADH production was 39%, while the glyoxylate cycle contributed up to 63% of the total NADH production on ethanol.

Next, we aimed at identifying how the flux increase through the glycoxylate cycle is achieved. Also here the Gibbs-free energies of reaction before and after glucose depletion provided important information. Interestingly, the two reactions in the glyoxylate cycle that utilize acetyl-CoA (malate synthase reaction, MALS, and the cytosolic citrate synthase reaction, CSc) are far from equilibrium at all analyzed time points (Figure 4C, reactions in red). The enzymes that catalyze these reactions are malate synthase 1 (Mls1p) and citrate synthase 2 (Cit2p), respectively. The limited catalytic activity of these two enzymes likely has a crucial role in regulating the distribution of acetyl-CoA between the mitochondria and the glyoxylate cycle. When glucose is depleted the expression of the Mls1p and Cit2p is strongly upregulated (Figure 4C, green arrows; Supplementary File 6), which likely draws most of the produced acetyl-CoA into the glyoxylate cycle. Even with the increased enzyme levels, the two reactions MALS and CSc remain far from equilibrium, which points to a remaining tight regulation of these reactions also after glucose depletion. Control of the CSc reaction, located at the branching point between the gluconeogenic reaction phosphoenolpyruvate carboxykinase (PPCK) and the glyoxylate cycle (see Figure 4C), probably also has an important role in rerouting part of the metabolic flux toward gluconeogenesis after glucose exhaustion. Thus, according to our results the concerted regulation at the protein expression level of both reactions, CSc and MALS, is a very important factor in increasing the flux through the glyoxylate cycle and redirecting part of that flux toward gluconeogenesis.

To reveal how the changes in protein expression leading to increased glyoxylate cycle flux are accomplished, we again performed the over-representation test taking into account the protein measurements before ( − 0.07 h) and after (2.02 and 5.13 h) glucose depletion. According to our analysis and in agreement with the literature (Haurie *et al*, 2001), the protein changes for fueling gluconeogenesis are primarily due to the activation of Cat8p (Figure 4D)—an important transcription factor involved in glucose de-repression—which occurs immediately after the switch of the carbon source. Later on, at 5.13 h, the transcription factors Hap4p and Sip4p are also activated (Figure 4D), which induce the glyoxylate cycle upregulation. The participation of Sip4p is limited to the regulation of only 23.5% of the genes that are upregulated at 5.13 h, while 70.6 and 88.2% of higher expressed proteins are under the control of Cat8p and Hap4p, respectively (Supplementary File 7).

In addition to these transcription factors that were known to be involved in the diauxic shift, our analysis also revealed transcription factors (Oaf1p, Abf1p, Yap5p and Azf1p; Figure 4D) that were not previously described to participate in the diauxic shift and that are activated simultaneously with Cat8p. Oaf1p was first described as a transcription factor responsive to oleate (Luo *et al*, 1996), but it was also reported to be recruited to the nucleus under low glucose conditions (Karpichev *et al*, 2008). While the mechanisms of the activation of Abf1p and Yap5p have not been elucidated, the former was reported to be involved in carbon source regulation (Brindle *et al*, 1990; Chambers *et al*, 1990; Yoo *et al*, 1995) and Yap5p activation requires a drop in PKA activity (Fernandes *et al*, 1997). Consistent with it being identified by our analysis, the expression of Azf1p was reported to be increased under non-fermentable growth conditions (Stein *et al*, 1998). Supporting our predictions, the deletion mutants of Oaf1, Yap5 and Azf1 (the ΔAbf1 mutant is not viable) all presented a marked defect of growth after glucose depletion (Supplementary Files 3.4 and 7).

### Changing the NADPH source

The last major adaptation event is a change in the way NADPH is regenerated (Figure 3D and E, 'III'). As revealed by the estimated flux ranges, during glucose consumption ( − 2.1 and − 0.6 h) and early ethanol consumption (0.8 and 4.4 h) the main source of NADPH is the oxidative branch of the PP pathway (as indicated by the flux through the 6-phosphogluconolactonase reaction, PGL, in Figure 3A–D). However, once the adaptation process is completed (7.4 h), the flux through the PP pathway is completely shut down (Figure 3E, cf. flux through 'PGL' and 'TKT1/TALA') and only the non-oxidative reactions synthesizing ribose and erythrose remain active (Figure 3E, 'TKT2'). Thus, our results are in agreement with previous studies, which report that (i) the PP pathway is required immediately after glucose depletion (Castelli *et al*, 2011; Yang *et al*, 2011), but (ii) it is not necessary to maintain growth on ethanol (Daran-Lapujade *et al*, 2004).

The Gibbs-free energies of the PP pathway reactions point to flux-affecting regulation at the PGL reaction, which presents far-from-equilibrium thermodynamics at all time points (Figure 3A–E; Supplementary File 5). The enzymes that

catalyze this reaction (Sol3p and Sol4p) were shown to be upregulated before and immediately after glucose depletion during diauxic shift experiments (Castelli *et al*, 2011), indicating that the regulation of the oxidative part of PP pathway (Figure 3A–E, 'PGL') might reside at the translational level. After glucose depletion, however, the concentrations of the PP pathway metabolites significantly drop (Figure 2C), and the resulting concentrations are such that 7.4 h after glucose depletion the transaldolase (TALA) and transketolase (TKT1) reactions also move away from equilibrium (Supplementary File 5), indicating active regulation of these reactions in the later stages of the adaptation. The concerted regulation of these three reactions (PGL, TKT1 and TALA) might be responsible for the redirection of the metabolic fluxes to only the reactions that lead to biomass precursors (i.e., erythrose and ribose) and the shutting down of the flux through the rest of the PP pathway.

While the NADPH production in the oxidative part of the PP pathway ceases, still NADPH is required for the biosynthetic processes and the question is where it is regenerated at the later stages of the adaption. Alternative possibilities for regeneration are the oxidative reactions of ethanol metabolism (i.e., through the oxidation of acetaldehyde to acetate; aldehyde dehydrogenase reaction, ALDD) and the glyoxylate cycle (i.e., through the oxidation of isocitrate to 2-oxoglutarate; ICDH reaction). Indeed, the measured abundances of the respective NADH- and NADPH-dependent enzymes at 7.4 h (Supplementary File 6) suggest a redistribution of fluxes toward NADPH generation in these reactions: (i) the isocitrate dehydrogenase cytoplasmic (Idp2p, the NADPH-dependent isoenzyme of ICDH) is strongly upregulated and reaches levels that are 24 times higher than in the glucose exponential phase ($-3.8$ h), and (ii) the aldehyde dehydrogenase 3 (Ald3p, an isoenzyme of ALDD that prefers NAD but can also utilize NADP as cofactor; Navarro-Avino *et al*, 1999) is also upregulated (Supplementary File 6). The concerted upregulation of these proteins likely increases NADPH production to substitute the PP pathway as the main source of this cofactor. Consistent with these results, it has been reported that a double mutant deficient on both the PP pathway and Idp2p,

loses viability when shifted to a medium with a non-fermentable carbon source (Minard and McAlister-Henn, 2001) and displays increased levels of NADP$^+$ (Minard and McAlister-Henn, 2005).

Even though the halted flux through the PP pathway occurs at some point between 4.4 and 7.4 h, the upregulation of Idp2p seems to take place gradually between glucose depletion and 5.1 h. This slow but steady increase in expression prevented us to identify which are the transcription factors that are specifically associated with the switch in NADPH source. However, the IDP2 gene has been shown to interact with transcription factors Cat8p, Oaf1p, Abf1p, Azf1p and Hap4p (Teixeira *et al*, 2006) that, according to our analysis, are activated earlier during the diauxic shift (Figure 4D). Thus, these transcription factors seem to be involved in the increased expression of NADP-dependent enzymes as well.

The switch in NADPH source might be a strategy for the cells to increase their energetic efficiency: under glycolytic conditions the utilization of the PP pathway to produce NADPH does not require investment of energy; however, under non-fermentative substrate conditions cells would need to increase the flux through gluconeogenesis (which is highly energy-demanding) to produce PP pathway precursors.

## Conclusions

Through our work, we untangled the temporal organization of the diauxic shift. As summarized in Figure 5, we found that the adaptation is prepared by a decrease in glycolytic flux caused by a rerouting of carbon flux toward reserve carbohydrates (Figure 5, 'I'), followed by the reversal of fluxes and the onset of the glyoxylate cycle (Figure 5, 'II'). Finally, we found that a ceased flux through the PP pathway is the last stage of the adaptation, occurring only 6 h after glucose depletion, which is accompanied by a change in NADPH source (Figure 5, 'III'). Moreover, we identify several reactions in the metabolic network (Figure 5, 'Regulatory site'), whose regulation most likely causes the observed changes in flux distribution (Figure 5, 'Fluxes') and the influence of metabolites levels

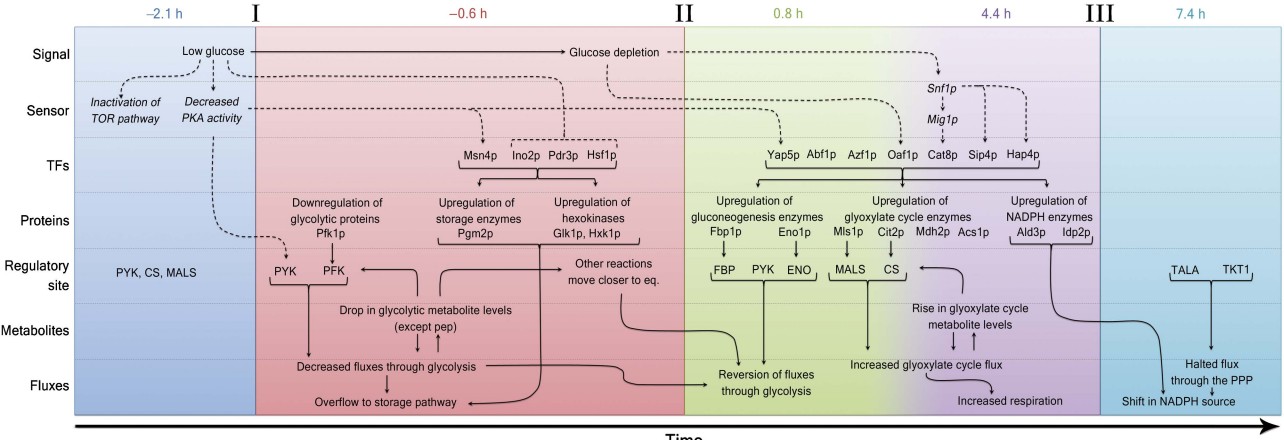

**Figure 5** Time-course reconstruction of the different events that lead to the adaptation based on the results presented in this report (regular text and solid arrows) and also supported by studies that were previously published (italics and dashed arrows) (Boy-Marcotte *et al*, 1996, 1998; DeRisi *et al*, 1997; Vincent and Carlson, 1998; Haurie *et al*, 2001; Haurie *et al*, 2004; Brauer *et al*, 2005; Radonjic *et al*, 2005; Slattery *et al*, 2008).

and protein abundances (Figure 5, 'Metabolites' and 'Proteins') on the observed flux distributions. Finally, we not only show the temporal activation of transcription factors that were already reported to participate in the diauxic shift, but also propose additional transcription factors that might be involved in achieving the observed changes in protein levels (Figure 5, 'TFs').

Apart from dissecting the temporal organization of the events that lead to the switch in carbon source we obtained some novel insights arising from the integrative analysis that challenge previous concepts. First of all, several studies have reported changes that occur before glucose depletion (DeRisi *et al*, 1997; Brauer *et al*, 2005; Radonjic *et al*, 2005), particularly the accumulation of storage carbohydrates (Lillie and Pringle, 1980; Parrou *et al*, 1999; Francois and Parrou, 2001) which is consistent with our results. However, while the accumulated glycogen was thought to be consumed after the switch in carbon source, our results suggest that there is no net reserve carbohydrate consumption at any time point after the switch. The usually observed decline in intracellular storage metabolite concentrations (Lillie and Pringle, 1980; Parrou *et al*, 1999; Francois and Parrou, 2001) is rather due to dilution of these compounds by cell growth and division. Thus, we propose that the redirection of flux toward storage compounds (before glucose depletion) is a consequence of the reduction in the flux through glycolysis.

Second, as it was also previously reported, after glucose exhaustion the glyoxylate cycle is activated. However, according to our results and contrary to previous preconceptions (Brauer *et al*, 2005), we found that, apart from its importance for generating the precursors for gluconeogenesis, the contribution of the glyoxylate cycle in energy production is even higher than that of the TCA cycle. We found that there is an upregulation at the translational level of the reactions that consume acetyl-CoA in the glyoxylate cycle, which causes a diversion fluxes toward this pathway.

Finally, there have been seemingly contradictory reports regarding the importance of the PP pathway during the diauxic shift. On the one hand, Castelli *et al* (2011) and Yang *et al* (2011) reported that the PP pathway is required for cell survival upon glucose exhaustion and, on the other hand, it has also been reported that this pathway is not necessary to maintain growth on ethanol (Daran-Lapujade *et al*, 2004). According to our results, the shutting down of the PP pathway is a very late event in the transition occurring only between 4.4 and 7.4 h after glucose depletion, a finding that could conciliate the previously reported results. Further, we found that there is an upregulation of NADP-utilizing enzymes whose expression is gradually increased during the first phases of the transition and that most likely replace the PP in NADPH regeneration once the transition is completed. The gradually increased expression of these enzymes (particularly Idp2p) explains why the shutting down of the flux through the PP pathway occurs only at the latest stage of the adaptation when the concentration of NADP-dependent enzymes is sufficiently high.

Besides the specific findings mentioned above, in this work we generated a unique data set on the diauxic shift in yeast comprising measurements of 69 intracellular metabolite concentrations, quantitative abundances of 72 enzymes of central metabolism and an accurate quantification of the physiological state acquired at a high time resolution to capture dynamic changes. While previously only transcriptome (DeRisi *et al*, 1997; Brauer *et al*, 2005; Radonjic *et al*, 2005) and comparative 2D electrophoresis proteome data (Boy-Marcotte *et al*, 1996; Boy-Marcotte *et al*, 1998; Vincent and Carlson, 1998; Haurie *et al*, 2001; Haurie *et al*, 2004) were available, the data presented here provide a detailed picture on the temporal organization of the diauxic shift spanning the different levels of cellular organization: from the activation of transcription factors to the overall physiological state of the cells, including the changes in protein levels, metabolite concentrations and metabolic flux distribution. Hence, this work represents the ground work for future studies toward generating a systems-level understanding about the diauxic shift, for example, generated through the development of kinetic models.

Furthermore, the general organization of the adaptation process is likely conserved in different species. In this sense, the present work could represent a great advantage in understanding how deregulation of the switch between glycolysis and gluconeogenesis can lead to metabolic diseases in humans, such as diabetes (Nuttall *et al*, 2008; Mather and Pollock, 2011).

## Materials and methods

### Strain used and culture conditions

*S. cerevisiae* FY4 (Winston *et al*, 1995) was grown in minimal defined medium with glucose as sole carbon source (Verduyn *et al*, 1992). In all, 50 ml precultures were inoculated from colonies taken from YPD plates and grown in 500 ml shake flasks at a glucose concentration of 10 g/l. A bioreactor (Bioengineering AG, Switzerland) with a working volume of 2 l minimal medium containing 5 g/l glucose was inoculated with 5 ml preculture from mid-exponential phase (initial $OD_{600}$ 0.01–0.02). The fermentation broth was stirred at 1200 r.p.m., aerated with 1 vvm air, and its temperature maintained at 30°C. The pH was controlled at a value of 5 by addition of 2 M KOH solution. To reduce ethanol evaporation, the offgas stream was led through a water cooled condenser and the precipitate recycled. If necessary, then a few drops of a 20% PEG 2000 solution were added as antifoaming agent. Three biologically independent cultures were carried out.

### Biomass and extracellular concentrations

Biomass concentrations were monitored by measuring optical density (OD) at 600 nm with a spectrophotometer (model Novaspec II, Pharmacia Biotech, USA). The correlation of OD with biomass dry weight (0.486 gDW/l OD) had previously been determined from a batch culture grown on 10 g/l glucose minimal medium. In these experiments, 15 ml of culture broth at five different ODs ranging from 0.3 to 3.3 were filtered using cellulose nitrate filters with a pore size of 0.45 µm (Sartorius AG, Germany) and the dried filters were weighed. To determine the extracellular concentrations of glucose, ethanol, acetate, pyruvate, succinate and glycerol, 1 ml samples were taken and centrifuged for 4 min at 4000 r.p.m. at 4°C. The supernatant was analyzed with an HPLC system (Agilent HP1100) as described by Heer and Sauer (2008), equipped with a polymer column (Aminex HPX-87H from BioRad, Switzerland). As eluent 5 mM $H_2SO_4$ was used and the column was heated to 60°C. The compounds were detected and quantified with a refractive index (RI) detector and an UV/Vis-detector (DAD). For absolute quantification, calibration curves with external standards for the corresponding pure substance obtained from Sigma (Switzerland) were used.

## Intracellular metabolite concentrations

For determination of intracellular metabolite concentrations, samples were withdrawn from the culture at specific time points. To quench metabolism and extract intracellular metabolites, the following procedure was applied (Ewald *et al*, 2009, modified from de Koning and van Dam (1992) and Gonzalez *et al* (1997)). Four samples of 1–4 ml at each sampling time point were quenched in 4 volumes of 60% methanol in 10 mM ammonium acetate pH 7.5 at −40°C. After centrifuging for 3 min at 14 000 r.p.m. and −9°C, they were frozen in liquid nitrogen and stored at −80°C. Intracellular metabolites were extracted by incubation in 75% ethanol 10 mM ammonium acetate pH 7.5 for 3 min at 95°C. The supernatant was retained by centrifuging at −9°C, samples were dried in a vacuum centrifuge, and two samples each were prepared for either LC-MS/MS of GC-TOF analysis.

For quantification by GC-TOF, two sample aliquots were derivatized with first methoxyamine solution (20 mg/ml methoxyamine hydrochloride (Supelco) in pyridine (analytical grade, Merck)) and then either TMS (*N*-methyl-*N*-(trimethylsilyl)-trifluoroacetamide (Fluka) or TBDMS *N*-tert-butyldimethylsilyl-*N*-methyltrifluoroacetamide with 1% *tert*-butyl-dimethylchlorosilane (Fluka). The samples were separated via GC on a HP5-MS (Hewlett-Packard, length 30 m × ID 0.25 × film 0.25 μm) column and injected (CIS4, Gerstel, Germany) for MS analysis to a TOF spectrometer (Pegasus III, Leco, Germany). Detailed information on process parameters are described in Ewald *et al* (2009) and Buscher *et al* (2009)). Leco ChromaTOF software (version 2.32) was used for machine control. An autosampler (MPS2, Gerstel, Germany, controlled by Gerstel Maestro software, version 1.2.3.5) was used for automatized derivatization and sample injection to the GC-TOF system.

For quantification by LC-MS/MS, an Ion-Pairing LC method adapted from Luo *et al* (2007) was applied (Buscher *et al*, 2009). The mobile phase was composed of eluent A (aqueous solution of 10 mM tributylamine and 15 mM acetic acid) and eluent B (methanol); the gradient profile was as follows: $t = 0$ min, 0% B; $t = 15$ min, 55% B; $t = 27$ min, 66% B; $t = 28$ min, 100% B. The end-capped C18 column Synergi Hydro RP, 2.1 ×150 mm, 4 μm particles (Phenomenex, Germany) was employed. The column was equilibrated for 20 min before each injection, the flow rate was 200 μl/min and the column temperature was controlled at 40°C. For tandem MS analysis a 4000 QTRAP linear ion trap mass Spectrometer (AB Sciex, Canada) was coupled to the LC. Analyst software (AB Sciex, Canada) was used for both machine control and data acquisition. All analyses were performed in negative ion and selected reaction monitoring mode with Q1 and Q3 set to unit resolution. Ion spray voltage, auxiliary gas temperature, nebulizer gas (GS1), auxiliary gas (GS2), curtain gas (CUR) and collision gas (CAD) were set to −4200 V, 650°C, 65, 40, 10, 4 (arbitrary units), respectively. Nitrogen (Pangas, Switzerland) was used as curtain and collision gas. Declustering potential (DP), collision energy (CE) and collision cell exit potential (CXP) were optimized separately for each transition. To obtain temporal resolution of >1 Hz for each transition, the run was divided into five segments and the dwell time for each transition was set to 50 ms.

## Protein concentrations

To determine protein concentration, duplicate samples of 25 ml were harvested from the same fermenter at −3.82, −1.7, −0.07, 2.02, 5.13 and 7.43 h relative to glucose exhaustion (P0–P5, respectively), put on ice and washed twice with 5 ml lysis buffer (cf. below). In between the washing steps, the protein samples were centrifuged for 5 min at 5000 r.p.m. at 4°C and the supernatant discarded afterwards. Cell pellets were frozen at −80°C.

For protein extraction, the cell pellets were thawed in an ice-cold lysis buffer containing 20 mM Hepes buffer pH 7.5, 2 mM DTT, 100 mM KCl, 10 mM EDTA, and complete yeast protease inhibitor cocktail (Roche, Germany), using 1 ml of lysis buffer per gram of yeast. Yeast cells were lysed by glass bead beating, and lysed cells were centrifuged to remove cellular debris. The supernatant was transferred to a fresh tube and the protein concentration in the extract was determined by Bradford assay. Proteins were precipitated by adding six volumes of cold (−20°C) acetone and resolubilized in a digestion buffer

containing 8 M urea and 0.1 M $NH_4HCO_3$. A 100-μg aliquot of each yeast protein sample was transferred to a fresh tube and mixed with an equal amount of [15]N-labeled yeast proteins (cf. below). Proteins were reduced with 12 mM DTT for 30 min at 35°C and alkylated with 40 mM iodoacetamide for 45 min at 25°C, in the dark. Samples were diluted with 0.1 M $NH_4HCO_3$ to a final concentration of 1.5 M urea and sequencing grade porcine trypsin (Promega, USA) was added to a final enzyme:substrate ratio of 1:100. The digestion was stopped by acidification with formic acid to a final pH of <3. Peptide mixtures were cleaned on Sep-Pak C18 cartridges (Waters, USA) eluted with 60% acetonitrile. Peptides were dried on a vacuum centrifuge, resolubilized in 0.1% formic acid and immediately analyzed.

Protein abundances were determined by a selective reaction monitoring (SRM)-based mass spectrometry approach as follows. Samples were analyzed on a hybrid triple quadrupole/ion trap mass spectrometer (4000QTrap, ABI/MDS-Sciex, Canada) equipped with a nanoelectrospray ion source. Chromatographic separations of peptides were performed on a Tempo nano LC system (Applied Biosystems, USA) coupled to a 15-cm fused silica emitter, 75 μm diameter, packed with a Magic C18 AQ 5 mm resin (Michrom BioResources, USA). Peptides were loaded on the column from a cooled (4°C) Tempo autosampler and separated with a linear gradient of acetonitrile/water, containing 0.1% formic acid, at a flow rate of 300 nl/min. A gradient from 5 to 30% acetonitrile in 30 or 60 min was used. For developing and validating the SRM assays, the mass spectrometer was operated in SRM mode, triggering acquisition of a full MS2 spectrum upon detection of an SRM trace (threshold 300 ion counts). SRM acquisition was performed with Q1 and Q3 operated at unit resolution (0.7 $m/z$ half maximum peak width) with an average of 100 transitions (dwell time 20 ms/transition) per run. MS2 spectra were acquired in the trap mode (enhanced product ion) for the two highest SRM transitions, using 100 ms fixed fill time, Q0 trapping enabled, Q1 resolution low, scan speed 4000 amu/s, $m/z$ range 300–1300, 2 scans summed. CEs for both SRM and MS2 analyses were calculated according to the formulas: $CE = 0.044 \times m/z + 5.5$ and $CE = 0.051 \times m/z + 0.5$ (CE: collision energy, $m/z$: mass-to-charge ratio of the precursor ion) for doubly and triply charged precursor ions, respectively.

For the quantitative analysis, a [15]N-labeled yeast digest was derived from a yeast batch culture that displayed diauxic growth on minimal medium with 20 g/l glucose and [15]N-labeled ammonium as nitrogen source. To gain high coverage of metabolism proteins, aliquots from the different phases (growth on glucose, transient phase and growth on ethanol) of this experiment were mixed. Such heavy labeled protein mixture was used as an internal standard for relative quantification and was spiked into each sample at a 1:1 (w:w) ratio before digestion. The three most intense transitions from each validated peptide and for the corresponding [15]N-labelled analogue were chosen for quantitative analysis in SRM mode. Quantitation was performed in scheduled-SRM mode, with Q1 and Q3 operated at unit resolution, using a cycle time of ∼3 s and a retention time window of 5 min. Peak height was determined with Multiquant 1.0.0.1 software (Applied Biosystems, USA/MDS-Sciex, Canada) after confirming for each peptide the co-elution of all six transitions.

The raw SRM data are publicly accessible through the PeptideAtlas database in http://www.peptideatlas.org/PASS/PASS00131.

## Calculation of specific uptake/excretion rates

The experimental data were fitted using the models indicated in Supplementary File 1, as described below. In the cases of OD, glucose, ethanol, glycerol and pyruvate, the data sets were split into two subsets—that is, before and after glucose exhaustion—and the fitting was done independently for each of these subsets. This was due to the inability of a unique continuous model to capture the discontinuity caused by the abrupt changes that occur upon glucose exhaustion.

For OD and glucose concentrations during the exponential phase, we used an exponential model as this best captures the underlying biology. For the rest of the experimental data sets, which do not follow a specific trend we performed a non-parametric regression approach using smoothing splines, in which the function is constructed on the basis of information derived from the data rather than being

predetermined. The best smoothing spline is obtained by minimizing the following criterion:

$$S(f) = \sum_{k=1}^{n} (Y_k - f(x_k))^2 + \lambda \int_{x_1}^{x_n} f''(x)^2 dx$$

where $(x_k, Y_k)$ is the series of experimental observations, $f$ the smoothing spline function and $\lambda$ the smoothing parameter. Although this method is a non-parametric regression, it does require the selection of $\lambda$, which determines the extent of smoothing and that is directly related to the effective degrees of freedom of the fitting. We calculated the optimal degrees of freedom in each case by performing a generalized cross-validation. Using the optimal degrees of freedom (with the minimum cross-validation score) sometimes led to splines that were not smooth enough and presented abrupt changes in their derivative. To avoid this, we used the lowest amount of degrees of freedom with a cross-validation score not higher than 103% of the minimum cross-validation score (see Supplementary File 1). The 95% confidence intervals for the smoothing splines and their derivatives were calculated according to Shipley *et al* (Shipley and Hunt, 1996). The experimental data, regression curves, degrees of freedom and residuals are provided as Supplementary data for each one of the metabolites measured and biomass (Supplementary File 1).

## Flux and Gibbs-free energy variability analysis

Maximal and minimal possible metabolic fluxes and possible Gibbs-free energy ranges for each reaction and during each phase were calculated using variability analysis. The model used is a 240-reaction stoichiometric model adapted from Jol *et al* (2012), which covers most of the central carbon and amino-acid metabolism taking into account compartmentalization into the cytoplasm and mitochondria. To this model, the reactions corresponding to storage compound production were added (see Supplementary File 2 for the complete model). In short, the above-determined extracellular fluxes were allowed to vary within their 95% confidence intervals subject to mass balances (except for protons, which were not included in the mass balance equations) and the second law of thermodynamics. The boundaries for measured metabolites were also set according to the 95% confidence intervals from the average of measurements between the following time points: $-8$ to $-1.5$ h, $-1.5$ to $0$ h, $0$ to $2$ h, $2$ to $6$ h and $6$ to $8$ h, and taking into account the compartmentalization of each metabolite. Specifically, for those metabolites that could be localized in more than one compartment, the 95% confidence interval upper limit was used to calculate the maximal concentration in each compartment (assuming that the metabolite is localized only in that compartment) and that value was set as the upper bound for the optimization. The ATP turnover (represented by ATP hydrolysis in the model) was also constrained to be at least 80% of the maximal possible ATP turnover considering the experimental rates and intracellular concentrations. For the exact limits that were actually used see Supplementary File 5.

The optimization problem was implemented as a mixed integer linear problem using GAMS and solved with the CPLEX solver, according to the following equations:

$$
\begin{aligned}
\min/\max \quad & v_j && \text{(for flux variability analysis) or}\\
& \Delta_r G_j && \text{(for Gibbs-free energy variability analysis)}\\
\text{subject to} \quad & \mathbf{S} \cdot \mathbf{v} = \mathbf{0}\\
& \Delta_r G_j = \Delta_r G_j'^0 + RT \sum_i S_{i,j} \ln c_i\\
& \Delta_r G_j v_j \leqslant 0\\
& v_j^{lo} \leqslant v_j \leqslant v_j^{up}\\
& v_{j,\exp} - \text{CI}_j^{95\%} \leqslant v_j \leqslant v_{j,\exp} + \text{CI}_j^{95\%}\\
& (\ln c_i)^{lo} \leqslant \ln c_i \leqslant (\ln c_i)^{up}\\
& \ln(c_{i,\exp} - \text{CI}_i^{95\%}) \leqslant \ln c_i \leqslant \ln(c_{i,\exp} + \text{CI}_i^{95\%})\\
& v_{\text{ATPS}} \geqslant 0.8 v_{\text{ATPS}}^{\max}
\end{aligned}
$$

where $v_j$ is the rate of each reaction, $\mathbf{S}$ is the stoichiometric matrix, $\Delta_r G_j$ is the Gibbs-free energy of reaction, $\Delta_r G_j'^0$ the transformed Gibbs-free energies of formation, $T$ is the temperature (298.15 K), $R$ is the gas constant, $c_i$ is the concentration of each reactant, $v_j^{lo}$ and $v_j^{up}$ are the lower and upper limits for the rates, $(\ln c_i)^{lo}$ and $(\ln c_i)^{up}$ are the lower and upper limits for the natural logarithm of the metabolite concentrations, $v_{j,\exp}$ is the experimentally determined rates and $\text{CI}_j^{95\%}$ the 95% confidence interval for those rates, $c_{i,\exp}$ is the experimental concentrations and $\text{CI}_i^{95\%}$ their confidence intervals, $v_{\text{ATPS}}$ is the rate of ATP hydrolysis and $v_{\text{ATPS}}^{\max}$ is the maximal possible rate of ATP hydrolysis with the previous constraints. See Supplementary File 5 for the exact bounds in each case.

## Hierarchical clustering

Proteins were clustered according to the similarity of their abundance changes during the five phases. For doing so, the fold changes with respect to P0 were used to compute the 'distance' value ($D$) for each pair of proteins, as defined by the equation $D = 1-r$, where $r$ is the Pearson product-moment correlation coefficient. The distance between different clusters was calculated using the average linkage method (Sokal and Michener, 1958). For each cluster and subcluster, a *P*-value was calculated taking into account the pathways to which the grouped proteins belong (according to the KEGG database) on the basis of a hypergeometric probability distribution. The complete clustering results including the *P*-values for each cluster and subcluster can be found in Supplementary File 6.

## Transcription factor analysis

Transcription factors, which are more often than by chance associated with the subset of proteins that were upregulated or downregulated between two adjacent measurement time points, were determined by a statistical analysis adopted from Boyle *et al* (2004). Subsets of upregulated and downregulated proteins were analyzed independently. For this analysis, we considered a protein to be regulated when the logarithm to base two of its fold-change with respect to the previous time point was at least twice the standard deviation. We used the transcription factor-gene associations (supported by both direct and indirect evidence) reported in the Yeastract database (Teixeira *et al*, 2006). For each transcription factor that is associated with the proteins for which concentrations were determined, a *P*-value based on a hypergeometric distribution was calculated.

$$p_j = 1 - \sum_{i=0}^{k_j-1} \frac{\dbinom{M_j}{i}\dbinom{N-M_j}{n-i}}{\dbinom{N}{n}}$$

Here, $N$ is the total number of proteins measured and $n$ the number of proteins that are upregulated or downregulated in each time point. $M_j$ and $k_j$ are the number of proteins that are related with a particular transcription factor, $j$, in the whole set of measured proteins and the subset of regulated ones, respectively. Such calculated *P*-value represents the probability that at least the observed number of associations with the subset of regulated proteins for a particular transcription factor would occur by chance.

## Supplementary information

## Acknowledgements

We would like to acknowledge funding from the EuroTango consortium for a fellowship for GZ, from YeastX and PhosphoNetX projects within the Swiss SystemsX.ch initiative, from UNICELLSYS, from the Netherlands Consortium for Systems Biology (Stimulus project) and from the Groningen Center of Synthetic Biology.

*Author contributions:* GZ performed the data analysis and integrated the different data sets. AK and Je planned the experiments and carried out the cultures and physiology measurements. BN contributed to the flux variability analysis; JE and NZ performed the metabolome measurements; PP performed the proteome measurements; SJ

analyzed the proteome data; RA supervised the proteomics work; US supported the work and helped with the writing. MH conceived the research. GZ and MH wrote the manuscript. All authors have read and approved the manuscript.

## Conflict of interest

The authors declare that they have no conflict of interest.

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
