## [Review Process File · Molecular Systems Biology]

Temporal system-level organization of the switch from glycolytic to gluconeogenic operation in yeast

Guillermo G. Zampar, Anne Kümmel, Jennifer Christina Ewald, Stefan Jol, Bastian Niebel, Paola Picotti, Ruedi Aebersold, Uwe Sauer, Nicola Zamboni, Matthias Heinemann

Corresponding author: Matthias Heinemann, University of Groningen

Review timeline:

Submission date:	15 August 2012
Editorial Decision:	10 October 2012
Revision received:	24 January 2013
Accepted:	21 February 2013

Editors: Andrew Hufton / Thomas Lemberger

Transaction Report:

1st Editorial Decision

10 October 2012

Thank you again for submitting your work to Molecular Systems Biology. We have now heard back from the three referees who agreed to evaluate your manuscript. As you will see below, the referees were cautiously supportive, but raised a series of important concerns, which we feel at sufficient to preclude publication of this work in its present form.

The editor would like to highlight two general, and somewhat related, issues.

First, the reviewers noted that while this work provided an integrated and detailed view of the diauxic shift, they noted that many of the specific findings were not necessarily novel. While they felt that this did not diminish the value of this work as a whole, the last reviewer in particular felt that greater effort was needed to highlight the novel findings, and possibly to test or validate in a more direct manner some of the most novel aspects.

Secondly, the Reviewer #2 raised substantial issues with the transcription factor analysis, and the physiological relevance of the most novel predictions. While this reviewer suggests reducing your claims or possibly even eliminating this analysis, the other reviewers and the editor feel that this is a crucial aspect of this work. As such, the issues raised by Reviewer #2 should be considered and addressed thoroughly, and where possible the most surprising or novel predictions should be further justified with existing data or new experiments.

In addition, when preparing your revised manuscript, please also deposit all new mass spectrometry data in a community supported public repository. Confidential reviewer logins should be provided with your revised work.

If you feel you can satisfactorily deal with these points and those listed by the referees, you may wish to submit a revised version of your manuscript. Please attach a covering letter giving details of the way in which you have handled each of the points raised by the referees. A revised manuscript will be once again subject to review and you probably understand that we can give you no guarantee at this stage that the eventual outcome will be favorable.

I would also like to acknowledge again that this review process has taken somewhat longer than usual, due to some initial delays in finding suitable reviewers. Please accept our apologies.

Referee reports:

Reviewer #1 (Remarks to the Author):

This manuscript by G.G. Zampar et al. provides a highly detailed view of a growth transition in the yeast *S. cerevisiae*. The diauxic shift, as it is, has attracted a lot of attention over decades of research and is still a great model for trying to understand how environmental changes can affect the choreography of complicated changes in cellular metabolisms and survival strategies. As such any efforts to improve on our current views of the diauxic shift by more precise and extensive data could be considered as interesting. Compared to previous studies the authors seem to do this with respect to measurements of intracellular metabolites and proteins. Their storyline relies heavily on the inclusion of transcriptional phenomena that are found in published accounts and data bases. However, they are able to build on this resource in some imaginative ways and are able to establish all together a coherent story. Also, since the extensive data collection presented in this work is based on state of the art methods executed with very high proficiency, I would definitely support publication of this manuscript.

There are, however, a few (mostly minor) points with respect to presentation and discussion that the authors might consider for improvement.

page 2-18 First sentence in introduction: to ensure homeostasis - of what ?

page 5-1 ... does not indicate a reduction in the growth and glucose uptake.

I would agree with the glucose uptake. Growth is a little bit more ambiguous - I would rather stay with the terminus "increase in biomass". It is clear from other observations (see the paper by Brauer et al., 2005) that cell size and budding clearly start to drop before glucose has run out and concomitant with an increase in protective measures such as glycogen production that some people would not consider as "growth". Since the rerouting of carbon flux is in the conclusion one might be more specific already in the results. Finally, considering Figure 1 A - does the curve not flatten out about half an hour before 0 glucose is attained - would be nice to see it at higher resolution?

page 5-18. Could one add a comment why the observation with PP is interesting?

Conclusion I am missing statements whether and where any differences have appeared between this study and previous studies.

Figure 1 Specific rate for biomass is not defined

Figure 2 I do not understand why panel D was included in this figure, besides that it is very difficult first of all to see and second to comprehend. The figure legend does not really help in my view.

page 20 make SRMs and MRMs consistent

Reviewer #2 (Remarks to the Author):

The paper by Zampar and coauthors is a study of the switch from glycolytic to gluconeogenic metabolism using *S. cerevisiae* as a model system. By combining measurements on biomass and extracellular metabolites with intracellular metabolomics and proteomics data, the authors provide a temporal picture of the yeast diauxic shift. Flux variability analysis is used to construct a model of the fluxes through the central carbon metabolism, and three distinct transitions are identified. The experiments are extended by a theoretical analysis where the YEASTRACT data base is used to search for candidate transcription factors involved in the diauxic shift.

In general, the experimental data are solid and believable. Many of the specific findings are of course not novel per se, but the study provides insight on the more subtle metabolic shifts and is a nice complement to the gene expression data that is available. The most interesting novel finding

that comes out of the study is the fact that activation of the glyoxalate cycle seems to be quantitatively more important than activation of the TCA cycle in the adaptation to growth on ethanol. While most of the other results already are known, the study provides a better resolution of the process by its careful measurements of metabolite and protein levels, and by its tentative identification through free energy calculations of certain key enzymes involved in the regulatory switch.

The weakest part of the study is the transcription factor analysis, which is likely to identify only some of the transcription factors involved in the diauxic shift. First, there is a bias in that the authors only look for transcription factors that may regulate the 50 enzymes identified as up- or downregulated in their proteomics study. This is a serious limitation. Second, the YEASTRACT data base is not up-to-date with respect to regulation of the diauxic shift where recent microarray studies have contributed to a better understanding of the process. Third, some important transcription factors are functionally duplicated, and will thus escape detection in data based only on single gene knockout studies. One example of this of relevance to the diauxic shift is Msn4 (identified in the present study) and Msn2 (missed in the present study) which are largely redundant. Another example is Gis1 and Rph1, which are partially redundant and also involved in the diauxic shift.

Furthermore, many genes are regulated both by the diauxic shift and in response to other signals. One example of this are proteins important for the stress response that are induced both by the diauxic shift and by heat shock. The YEASTRACT data base includes data on transcription factor-gene interactions which do not address in what context these interactions occur. The unexpected finding in the present study that the heat shock transcription factor Hsf1 is involved in the diauxic shift is therefore likely to be incorrect. The same may well be true for other "novel" diauxic shift transcription factors identified in the present study, i.e. Pdr3, Ino2, Oaf1, Abf1, Yap5 and Azf1.

Given these problems, the transcription factor study, if included at all, should be carefully hedged with statements to the effect that important transcription factors may have been missed, and that some of the novel transcription factors found may not in fact be involved in the diauxic shift but rather regulate these target genes under completely different conditions. Alternatively, the transcription factor study could be excluded from the paper since the other more interesting results stand on their own.

Minor points

- 1) The last sentence of the second paragraph in the Introduction reads as if the TOR pathway is activated at the diauxic shift. In fact, it is the opposite; TOR is active under nutrient rich conditions, and then inactivated upon nutrient limitation (in many regards rapamycin treatment has the same effect as the diauxic shift). Likewise, Figure 5 could be interpreted to mean that TOR (in contrast to PKA) is activated by lowered amounts of glucose, when it should instead be the opposite.
- 2) There are references to Figure 4C on page 9 line 17 and page 12 line 21 that I believe should be 4D.
- 3) On page 12 lines 14 and 16 there are two references to a Figure 6A - such a figure does not exist.
- 4) On page 6 line 10 it says that: "the concentrations of intracellular metabolites are nearly constant at the intermediate time points as well (Fig. 2A-C and Supplement 2)." This seems to be the case for the glycolytic and PPP intermediates, but not the TCA/glyoxalate cycle (Fig. 2B).
- 5) On page 11 line 20 it says: "The enzymes that catalyze these reactions are malate synthase 1, MASY, and CISO2, respectively." It should be clarified that we are talking about two and not three enzymes, i.e.: "malate synthase 1 (MASY) and citrate synthase 2 (CISO2)". It would be even better to use the standard gene/protein names here to avoid all confusions, i.e. Mls1 and Cit2. This is true also for the other enzymes mentioned in the text and figures - the standard names are preferable since they always identify exactly what genes or proteins are referred to.
- 6) At the top of page 16 the authors point out their proteomics data are more accurate than previous studies of the transcriptome and proteome during the diauxic shift. This may be true, but it would

still be useful to know how their new data compare to those of previous studies for the 72 proteins under investigation. Such a comparison should be included as a supplement and commented briefly on in the text.

Reviewer #3 (Remarks to the Author):

Review of "Temporal system-level organization of the switch from glycolytic to gluconeogenic operation in yeast" by Zampar et al.

In this manuscript, the authors present a coherent description of the temporal changes in metabolism that take place during the glucose to ethanol diauxic shift in *S. cerevisiae*.

Perhaps the most original aspect of this work on a non-original problem is the use of an impressive suite of advanced experimental techniques, modeling and analysis approaches as well as theoretical considerations. The result is a scientifically satisfying high quality work.

They found that there are three phases. First, when glucose levels drop, there is a reduction in the flux through glycolysis, in which some of the carbon is diverted to storage. Then, when glucose is finally depleted, there is a reversal in the C flow of glycolysis with the activation of the glyoxylate cycle (GC), an activation higher than that of the TCA cycle. Here, the authors predict that GC would be a more important source of NADH+ than TCA. Finally, in a third phase, the pentose phosphate cycle shuts off and NADPH+ shifts to be made by induced isoenzymes that use NADP+ as cofactor.

In addition to this temporal description metabolic adjustments, they also provide good evidence as to the points of control responsible for the transitions from one phase to the next, and to the regulatory strategies employed by the cell in each case. They were able to determine in most cases if regulation was at the post-translational or gene expression level. In the latter, they predicted the transcription factors involved.

They measured metabolites by GC-TOF and LC-TOF. Because they couldn't measure experimentally the intracellular fluxes, they estimated possible flux ranges using flux variability analysis. For this, they used a big stoichiometric model of yeast metabolism constrained by the uptake and excretion rates and the measured metabolites, which they smartly coupled to the model using thermodynamic considerations. In fact, a substantial part of their conclusions derive from Gibbs energy analysis.

Abundance of proteins was determined by "selective reaction monitoring" based mass spec. And the transcription factors putatively involved in the different phases of the diauxic shift were established using an approach that takes into considerations the measured changes in protein abundance between consecutive time points and published data on genes induced by yeast transcription factors.

Critique:

1- Given that diauxic shift has been studied for years, many of the findings presented here have been previously reported (as the authors mention throughout the text). It would greatly serve the manuscript a more thorough comparison of past knowledge and new discoveries, especially focusing on the system-level perspective provided by the current approach. This would help answer the question of what is really new, in the sense that it couldn't have been proposed based on reading the literature.

2- Page 4-5. I see the change in CO₂ production starting at -90 min, but I don't see a simultaneous reduction in ethanol, glycerol and succinate as the authors indicate. Could the authors clarify where should we look?

3- Page 8. It is not clear if the authors have actually measured an increase in storage carbohydrates or their conclusion is inferred from the analysis of the data coupled to the metabolic model. If not, measuring glycogen would strengthen one of their main conclusions. Here is a good example of my point number 1. In page 9 it says that it has already been reported an increase in glycogen during diauxic shift. Thus, have the authors re-discovered it? What is new here?

4- Figure 2D and 3F are hard to see and interpret.

5- I am not an expert in metabolism, but there seem to be a confusion in gene naming and/or co-factor dependencies. According to the SGD, the ALDH3 (ALD3, YMR169c) gene referred to by the authors in the NAPH+ section (page 14) codes for an Aldehyde dehydrogenase enzyme that is NAD+ dependent instead of NADP+ dependent as the authors claim and state in the main text and Supplement 6.

Similarly, in the same Supplement 6, ALD5 (a K+ activated, NADP+ dependent enzyme according to SGD) appears as a K+ independent enzyme; and ALD4 (a K+ activated, NAD+ or NADP+ dependent enzyme according to SGD) appears as a K+ independent NAD+ using enzyme.

Please clarify this issue and explain how it affects their conclusions regarding NADPH+ generation.

6- The authors do a god job in explaining the evidence and reasoning behind the analysis throughout the manuscript. They show how they go from metabolites to fluxes, to key regulated reactions, to proteins involved, to transcription factors (regulatory networks). However, for the third phase, they apparently have not determined the putative transcription factors involved in the selective upregulation of the NADP+ dependent IDHC and ALDH enzymes. Was it not possible? Could they do it?

1st Revision - authors' response

24 January 2013

First, the reviewers noted that while this work provided an integrated and detailed view of the diauxic shift, they noted that many of the specific findings were not necessarily novel. While they felt that this did not diminish the value of this work as a whole, the last reviewer in particular felt that greater effort was needed to highlight the novel findings, and possibly to test or validate in a more direct manner some of the most novel aspects.

As suggested by the reviewers and editor, we added three paragraphs in the conclusions section that highlight the most important and novel findings of our work.

Secondly, the Reviewer #2 raised substantial issues with the transcription factor analysis, and the physiological relevance of the most novel predictions. While this reviewer suggests reducing your claims or possibly even eliminating this analysis, the other reviewers and the editor feel that this is a crucial aspect of this work. As such, the issues raised by Reviewer #2 should be considered and addressed thoroughly, and where possible the most surprising or novel predictions should be further justified with existing data or new experiments.

The predictions of the possibly involved transcription factors can be supported by previous results reported in the literature. For example, we predicted Yap5p to be involved in the diauxic shift. It has been reported that Yap5p activation requires a drop in PKA activity, which, interestingly, also occurs prior to glucose depletion in the diauxic shift. Likewise, Oaf1p, another newly predicted transcription factor to be involved in the diauxic shift, was reported to be recruited to the nucleus under low glucose conditions so its involvement during the diauxic shift is also very likely. Considerations like these ones regarding every predicted transcription factor, along with the respective references, are clearly stated in the main text of the manuscript.

For this revised manuscript, in order to get additional experimental evidence supporting the involvement of the predicted transcription factors involved in the diauxic shift, we constructed deletion mutants of each one of them and analyzed their ability to grow throughout the diauxic shift. Here, we found that these deletion mutants show significant defect in growth rate upon glucose depletion as compared with the wild type, which strongly supports our predictions. We added this additional evidence to the manuscript (see Supplement 3.4 and Supplement 7) and we also added a remark stating that for an ultimate proof of the involvement further follow up experiments would be necessary.

In addition, when preparing your revised manuscript, please address the following format and content issues:

1. Please deposit all new mass spectrometry data in a community supported public repository. Confidential reviewer logins should be provided with your revised work.

The mass spectrometry data have been deposited in the PeptideAtlas database (<http://www.peptideatlas.org>) and it can be accessed using the following credentials:

Dataset Identifier: PASS00131

Password: QF2869zq

2. The quality of the current figure images is somewhat low, and the text is noticeably blocky/blurry. You will get the best results if the figures are remade in a professional quality vector graphics program, like Illustrator or Inkscape, and saved directly in the EPS or PDF formats.

All the images accompanying the resubmission have been remade in order to meet the quality standards.

3. Please provide three to four 'bullet points' highlighting the main findings of your study.

4. Please provide a 'standfirst text' summarizing the study in one or two sentences (approx. 250 characters).

The bullet points describing the main findings and the standfirst text are included in the submission.

5. Please provide a "thumbnail image" (width=211 x height=157 pixels, jpeg format), which can be used to highlight your paper on our homepage.

As requested, a thumbnail image is now submitted along with the revised version of the manuscript.

Reviewer #1 (Remarks to the Author):

*This manuscript by G.G. Zampar et al. provides a highly detailed view of a growth transition in the yeast *S. cerevisiae*. The diauxic shift, as it is, has attracted a lot of attention over decades of research and is still a great model for trying to understand how environmental changes can affect the choreography of complicated changes in cellular metabolisms and survival strategies. As such any efforts to improve on our current views of the diauxic shift by more precise and extensive data could be considered as interesting. Compared to previous studies the authors seem to do this with respect to measurements of intracellular metabolites and proteins. Their storyline relies heavily on the inclusion of transcriptional phenomena that are found in published accounts and data bases. However, they are able to build on this resource in some imaginative ways and are able to establish all together a coherent story. Also, since the extensive data collection presented in this work is based on state of the art methods executed with very high proficiency, I would definitely support publication of this manuscript.*

There are, however, a few (mostly minor) points with respect to presentation and discussion that the authors might consider for improvement.

page2-18 First sentence in introduction: to ensure homeostasis - of what ?

Indeed, the word “homeostasis” is probably not the best option in the case of the diauxic shift since there is a drastic change in the homeostatic state of the cells when glucose is replaced by ethanol as the main carbon source. As suggested by the reviewer, we reconsidered this first sentence of the introduction and replaced the word “homeostasis” with “survival”. We believe that the new phrasing reflects the fact that cells adapt their metabolic state in order to continue growing under different conditions.

page 5-1 ... does not indicate a reduction in the growth and glucose uptake. I would agree with the glucose uptake. Growth is a little bit more ambiguous - I would rather stay with the terminus "increase in biomass". It is clear from other observations (see the paper by Brauer et al., 2005) that cell size and budding clearly start to drop before glucose has run out and concomitant with an increase in protective measures such as glycogen production that some people would not consider as "growth". Since the rerouting of carbon flux is in the conclusion one might be more specific already in the results. Finally, considering Figure 1 A - does the curve not flatten out about half an hour before 0 glucose is attained - would be nice to see it at higher resolution?

The point arisen by the reviewer is totally valid and it was not taken into account in the previous version of the manuscript. We now use the expression "rate of biomass production" instead of "growth rate" to acknowledge all processes that generate biomass and not only the production of daughter cells.

Regarding a potential flattening out of the biomass curve, some authors have reported a decrease in the rate of biomass production before glucose depletion; however, on the basis of our results such flattening out is not apparent. The data set that was obtained in our experiments consists of 14 data points between -0.53 and 0 h and 24 data points between -1 and 0 h, which is considerably of higher resolution than previous studies. As it can be seen in Supplement 1 - Biomass, only 5 out of the 14 data points between -0.53 and 0 h have negative residuals and the average residual in this range is positive (0.0126 gDW/l). This clearly shows that the exponential fit is underestimating the experimental measurements rather than overestimating them, so there is no reason to envision a flattening out of the biomass curve according to our data. In any case, it is possible that the biomass production slows down right before glucose exhaustion but it would require a very detailed analysis in this time range in order to get clear conclusions.

page 5-18. Could one add a comment why the observation with PP is interesting?

(We assume that the reviewer meant "PEP" instead of "PP"). The accumulation of PEP arises, according to our data, as a consequence of the restriction of fluxes through the pyruvate kinase reaction (PYK). However, as this comes later in the text (because it requires the FVA results), adding a comment in this section would seem out of place. Instead we added a reference to the section in which these changes are discussed.

Conclusion I am missing statements whether and where any differences have appeared between this study and previous studies.

A summary of the most interesting and novel findings was added in the Conclusions section as suggested by the reviewer.

Figure 1 Specific rate for biomass is not defined

As noticed by the reviewer, the specific rate of biomass production was not defined in the original version of the manuscript. We have amended the legend to Figure 1 in order to include the calculation of this specific rate. The legend now reads: "The regression curves were derived and divided by the cellular dry weight at each time point to calculate the specific rate of biomass production and the specific rates of uptake/excretion of each metabolite (shown in blue)".

Figure 2 I do not understand why panel D was included in this figure, besides that it is very difficult first of all to see and second to comprehend. The figure legend does not really help in my view.

The original purpose of Fig. 2D was to show the ranges of Gibbs free energies of the reactions that are close and far from equilibrium (those highlighted in Fig. 3A-E). However, we agree with the reviewer that it is not easy to comprehend and it is not really necessary for understanding the main story of the manuscript. Because of this we decided to take it out from the main text in order not to draw attention on details. In any case, all the information about the Gibbs energies of reaction is accessible to readers in Supplement 5. ("Results (free energies)" and "Plots" tabs).

page 20 make SRMs and MRMs consistent

The text was corrected as suggested by the reviewer.

Reviewer #2 (Remarks to the Author):

*The paper by Zampar and coauthors is a study of the switch from glycolytic to gluconeogenic metabolism using *S. cerevisiae* as a model system. By combining measurements on biomass and extracellular metabolites with intracellular metabolomics and proteomics data, the authors provide a temporal picture of the yeast diauxic shift. Flux variability analysis is used to construct a model of the fluxes through the central carbon metabolism, and three distinct transitions are identified. The experiments are extended by a theoretical analysis where the YEASTRACT data base is used to search for candidate transcription factors involved in the diauxic shift.*

In general, the experimental data are solid and believable. Many of the specific findings are of course not novel per se, but the study provides insight on the more subtle metabolic shifts and is a nice complement to the gene expression data that is available. The most interesting novel finding that comes out of the study is the fact that activation of the glyoxalate cycle seems to be quantitatively more important than activation of the TCA cycle in the adaptation to growth on ethanol. While most of the other results already are known, the study provides a better resolution of the process by its careful measurements of metabolite and protein levels, and by its tentative identification through free energy calculations of certain key enzymes involved in the regulatory switch.

*The weakest part of the study is the transcription factor analysis, which is likely to identify only some of the transcription factors involved in the diauxic shift. First, there is a bias in that the authors only look for transcription factors that may regulate the 50 enzymes identified as up- or downregulated in their proteomics study. This is a serious limitation. Second, the YEASTRACT data base is not up-to-date with respect to regulation of the diauxic shift where recent microarray studies have contributed to a better understanding of the process. Third, some important transcription factors are functionally duplicated, and will thus escape detection in data based only on single gene knockout studies. One example of this of relevance to the diauxic shift is *Msn4* (identified in the present study) and *Msn2* (missed in the present study) which are largely redundant. Another example is *Gis1* and *Rph1*, which are partially redundant and also involved in the diauxic shift.*

As pointed out by the reviewer, the list of transcription factors predicted to be involved in the diauxic shift is far from complete and by no means we claim so. We are aware that with our analysis only transcription factors related to the measured proteins can be identified. However, this is not a drawback because we are especially interested in the transcription factors that realize the changes in abundances of enzymes in the central carbon metabolism. Discovering all transcription factors involved in the diauxic shift goes beyond the scope of this study and would require a completely different approach. Likewise, the absence of some gene-transcription factor interactions in the Yeabstract database (either due to it not being up-to-date or because of functional redundancies) would result in some transcription factors not being identified but it does not in any way invalidate the novel transcription factors predicted to participate in the diauxic shift (see below). So, the intention of the over-representation analysis is to propose novel transcription factors that could be activated during the diauxic shift and that would accomplish the observed changes in the abundance of the measured proteins (central metabolism enzymes), rather than to provide a complete list of all involved transcription factors.

*Furthermore, many genes are regulated both by the diauxic shift and in response to other signals. One example of this are proteins important for the stress response that are induced both by the diauxic shift and by heat shock. The YEASTRACT database includes data on transcription factor-gene interactions which do not address in what context these interactions occur. The unexpected finding in the present study that the heat shock transcription factor *Hsf1* is involved in the diauxic shift is therefore likely to be incorrect. The same may well be true for other "novel" diauxic shift transcription factors identified in the present study, i.e. *Pdr3*, *Ino2*, *Oaf1*, *Abf1*, *Yap5* and *Azf1*.*

Although it may seem surprising at first, it should be kept in mind that it is not uncommon that proteins have different (and unexpected) functions and that they can be activated (or repressed) by

alternative pathways. To identify transcription factors that are likely also involved in the diauxic shift (besides being involved in their “classical” function) we precisely needed an approach – such as used the over-representation analysis – that would not make use of any information about the context of the known interaction!

In fact please note that the results of our predictions are in most cases actually not that far-fetched: Previous reports support the link of the predicted transcription factors with the diauxic shift. For example, it has been reported under different conditions that Yap5p is activated by a drop in PKA activity, which also occurs upon glucose exhaustion in the diauxic shift (see main text for a brief discussion on the other transcription factors).

Nevertheless, to add further support to our predictions, we constructed deletion mutants for all the predicted transcription factors and analyzed their growth during the diauxic shift (Supplement 3.4 and Supplement 7). Interestingly, the disruption of the Hsf1 gene rendered cells inviable even under non-stressed conditions, which indicates that this transcription factor is essential to cellular processes other than heat-shock survival. Likewise, Pdr3, Oaf1, Yap5 and Azf1 deletions revealed a clear detrimental effect on the ability of cells to adapt to the shift in carbon source, and the Ino2 deletion mutant showed a markedly reduced growth rate even on glucose (see Supplement 3.4 and Supplement 7). The deletion mutant of Abf1 was also found to be inviable. These experiments add further support to our predictions. However, to ultimately prove their involvement further experimental analysis would be necessary. We clearly acknowledge this in the revised manuscript.

Given these problems, the transcription factor study, if included at all, should be carefully hedged with statements to the effect that important transcription factors may have been missed, and that some of the novel transcription factors found may not in fact be involved in the diauxic shift but rather regulate these target genes under completely different conditions. Alternatively, the transcription factor study could be excluded from the paper since the other more interesting results stand on their own.

As suggested by the reviewer, we chose to keep the transcription factor analysis in the manuscript but we clearly state that it should be regarded as a prediction only and not as a proven fact.

Minor points

1) The last sentence of the second paragraph in the Introduction reads as if the TOR pathway is activated at the diauxic shift. In fact, it is the opposite; TOR is active under nutrient rich conditions, and then inactivated upon nutrient limitation (in many regards rapamycin treatment has the same effect as the diauxic shift). Likewise, Figure 5 could be interpreted to mean that TOR (in contrast to PKA) is activated by lowered amounts of glucose, when it should instead be the opposite.

As noticed by the reviewer, the statement about the TOR pathway was not correct in the original manuscript. We amended the text and Fig. 5 to clarify this point.

2) There are references to Figure 4C on page 9 line 17 and page 12 line 21 that I believe should be 4D.

3) On page 12 lines 14 and 16 there are two references to a Figure 6A - such a figure does not exist.

The references to figures pointed out by the reviewer were indeed incorrect. We changed the wrong references so they point to the correct figures.

4) On page 6 line 10 it says that: "the concentrations of intracellular metabolites are nearly constant at the intermediate time points as well (Fig. 2A-C and Supplement 2)." This seems to be the case for the glycolytic and PPP intermediates, but not the TCA/glyoxalate cycle (Fig. 2B).

As the reviewer commented, it is true that for the TCA/glyoxalate cycle intermediates the concentrations are not constant for a large period of time, thus the FVA would not be valid. However, precisely for this reason we chose time points where the metabolite concentrations remain as constant as possible. Please note that time point 0.8 h is right before the concentrations of the TCA/glyoxalate cycle intermediates start increasing and time point 4.4 h is after the concentrations

are (to some extent) stabilized again. We avoided using time points in between because in this range the rate of formation of metabolites of the TCA/glyoxylate cycle is clearly not zero.

5) On page 11 line 20 it says: "The enzymes that catalyze these reactions are malate synthase 1, MASY, and CISOY2, respectively." It should be clarified that we are talking about two and not three enzymes, i.e.: "malate synthase 1 (MASY) and citrate synthase 2 (CISOY2)". It would be even better to use the standard gene/protein names here to avoid all confusions, i.e. Mls1 and Cit2. This is true also for the other enzymes mentioned in the text and figures - the standard names are preferable since they always identify exactly what genes or proteins are referred to.

As suggested by the reviewer, in the new version of the manuscript we refer to all enzymes using their gene names. Supplement 6 was modified accordingly.

6) At the top of page 16 the authors point out their proteomics data are more accurate than previous studies of the transcriptome and proteome during the diauxic shift. This may be true, but it would still be useful to know how their new data compare to those of previous studies for the 72 proteins under investigation. Such a comparison should be included as a supplement and commented briefly on in the text.

Haurie *et al.* (2001) and Haurie *et al.* (2004) experiments were designed to elucidate the participation of different transcription factors (Cat8, Sip4) during the diauxic shift. In these experiments, the authors culture yeast cells and radioactively label newly synthesized proteins 15 min after glucose depletion. The samples are then analyzed by means of 2D gel electrophoresis. The authors perform this analysis using a WT strain and deletion mutants of the transcription factor under study and compare the expression patterns in both cases. Because of the experimental setup that is used in these experiments, all proteins that are synthesized after glucose depletion are labeled regardless their participation in the diauxic shift. As such, they don't get information about the differences in protein abundances throughout the adaptation period but rather they are only able to identify the proteins whose expression is dependent on the transcription factor under study. That is why we call these experiments "less quantitative" (maybe "qualitative" or "comparative" would have been more accurate). In contrast, in our experiments we are able to assess the changes in abundances of each enzyme by measuring protein abundances at different time points during the transition. Because of the different kind of data obtained from our and previous experiments, it is not possible to make a direct comparison; they simply reveal different aspects of the shift. This was probably not clear in the previous version of the manuscript so we modified the text to explain it better.

Reviewer #3 (Remarks to the Author):

*Review of "Temporal system-level organization of the switch from glycolytic to gluconeogenic operation in yeast" by Zampar *et al.**

*In this manuscript, the authors present a coherent description of the temporal changes in metabolism that take place during the glucose to ethanol diauxic shift in *S. cerevisiae*.*

Perhaps the most original aspect of this work on a non-original problem is the use of an impressive suite of advanced experimental techniques, modeling and analysis approaches as well as theoretical considerations. The result is a scientifically satisfying high quality work.

They found that there are three phases. First, when glucose levels drop, there is a reduction in the flux through glycolysis, in which some of the carbon is diverted to storage. Then, when glucose is finally depleted, there is a reversal in the C flow of glycolysis with the activation of the glyoxylate cycle (GC), an activation higher than that of the TCA cycle. Here, the authors predict that GC would be a more important source of NADH⁺ than TCA. Finally, in a third phase, the pentose phosphate cycle shuts off and NADPH⁺ shifts to be made by induced isoenzymes that use NADP⁺ as cofactor.

In addition to this temporal description metabolic adjustments, they also provide good evidence as to the points of control responsible for the transitions from one phase to the next, and to the

regulatory strategies employed by the cell in each case. They were able to determine in most cases if regulation was at the post-translational or gene expression level. In the latter, they predicted the transcription factors involved.

They measured metabolites by GC-TOF and LC-TOF. Because they couldn't measure experimentally the intracellular fluxes, they estimated possible flux ranges using flux variability analysis. For this, they used a big stoichiometric model of yeast metabolism constrained by the uptake and excretion rates and the measured metabolites, which they smartly coupled to the model using thermodynamic considerations. In fact, a substantial part of their conclusions derive from Gibbs energy analysis.

Abundance of proteins was determined by "selective reaction monitoring" based mass spec. And the transcription factors putatively involved in the different phases of the diauxic shift were established using an approach that takes into considerations the measured changes in protein abundance between consecutive time points and published data on genes induced by yeast transcription factors.

Critique:

1- Given that diauxic shift has been studied for years, many of the findings presented here have been previously reported (as the authors mention throughout the text). It would greatly serve the manuscript a more thorough comparison of past knowledge and new discoveries, especially focusing on the system-level perspective provided by the current approach. This would help answer the question of what is really new, in the sense that it couldn't have been proposed based on reading the literature.

As suggested by the reviewer, we included a summary of our results in the Conclusions section in order to highlight what is novel about them.

2- Page 4-5. I see the change in CO₂ production starting at -90 min, but I don't see a simultaneous reduction in ethanol, glycerol and succinate as the authors indicate. Could the authors clarify where should we look?

Even though the decrease in the specific rates of excretion of ethanol, succinate and glycerol is less steep than that of carbon dioxide production, there is a drop of around 20% in the former specific rates when time points -2.1 h and -0.6 h. This decrease can be seen in the blue curves of Fig. 1E, D and F respectively. We now included in the main text the references to the specific figures and the percentages of decrease in each case.

3- Page 8. It is not clear if the authors have actually measured an increase in storage carbohydrates or their conclusion is inferred from the analysis of the data coupled to the metabolic model. If not, measuring glycogen would strengthen one of their main conclusions. Here is a good example of my point number 1. In page 9 it says that it has already been reported an increase in glycogen during diauxic shift. Thus, have the authors re-discovered it? What is new here?

Indeed, we have not measured glycogen cellular content mainly because of the very detailed measurements that were done previously by other authors (Lillie *et al.* 1980; Parrou *et al.* 1999; Francois *et al.* 2001). Instead, the estimation of the fluxes towards the formation of storage compounds comes as a consequence of the FVA (see Supplement 3.2). In this sense, our estimations are consistent with the previously reported measurements (accumulation of storage compounds right before glucose depletion), which supports our method. The novel findings in this regard is that there is no net consumption of reserve carbohydrates after the switch in carbon source (i.e. $t = 0.8$ h and $t = 4.4$ h) which strongly suggests that, contrary to what is generally believed, glycogen is not accumulated for its later utilization as a carbon source. Instead, its accumulation seems to be a consequence of the limitation in the glycolytic flux before glucose depletion. In order to keep the main text concise, these conclusions were presented in Supplement 3.2 of the previous version of the manuscript. We have now included them in the main text also.

4- Figure 2D and 3F are hard to see and interpret.

Regarding Fig. 2D, we decided to take it out of the main text because, as the reviewer states, it is not easy to interpret and it does not add essential information for understanding the overall conclusions (see answer to Reviewer #1). However, we consider that Fig. 3F is necessary because it highlights the fact that the results from the FVA are not exact fluxes but rather ranges of possible fluxes. In this sense, Fig. 3A-E can be misleading to the readers if left alone. Please note that the fluxes indicated by the arrows in Fig. 3A-E are averages of the maximum and minimum values of the ranges obtained by the FVA and by no means represent exact values. Although less graphically appealing, Panel F contains the actual results of the FVA (the flux ranges) and we consider it important to keep it in the main text so readers can verify that our conclusions hold even when we don't obtain unique flux values. We added a paragraph in the main text to clarify this point.

5- I am not an expert in metabolism, but there seem to be a confusion in gene naming and/or cofactor dependencies. According to the SGD, the ALDH3 (ALD3, YMR169c) gene referred to by the authors in the NAPH+ section (page 14) codes for an Aldehyde dehydrogenase enzyme that is NAD+ dependent instead of NADP+ dependent as the authors claim and state in the main text and Supplement 6.

Similarly, in the same Supplement 6, ALD5 (a K+ activated, NADP+ dependent enzyme according to SGD) appears as a K+ independent enzyme; and ALD4 (a K+ activated, NAD+ or NADP+ dependent enzyme according to SGD) appears as a K+ independent NAD+ using enzyme. Please clarify this issue and explain how it affects their conclusions regarding NADPH+ generation.

Indeed, as noted by the reviewer, the aldehyde dehydrogenase Ald3p is annotated in SGD as a NAD+ specific enzyme; however, as described by Navarro-Aviño *et al.* (1999) this enzyme can also use NADP+ as cofactor, although less efficiently. We hypothesize that, even when NAD+ is the major cofactor for Ald3p, its up-regulation might also contribute to the regeneration of NADPH. This point was probably not clear in the previous version of the manuscript so we have modified the text in order to acknowledge this fact.

Regarding the other aldehyde dehydrogenases, there are indeed some discrepancies regarding NAD+/NADP+ dependencies between the SGD and Uniprot databases. For example, Ald4p is annotated as NAD+ dependent in the latter while it is referred as NADP+ dependent in SGD. In order to solve these differences we consulted the original papers and made the appropriate changes in Supplement 6. Please note that our conclusions still hold because the abundances of these enzymes decrease (or remain constant) so it is very unlikely that they are involved in NADPH regeneration.

6- The authors do a good job in explaining the evidence and reasoning behind the analysis throughout the manuscript. They show how they go from metabolites to fluxes, to key regulated reactions, to proteins involved, to transcription factors (regulatory networks). However, for the third phase, they apparently have not determined the putative transcription factors involved in the selective upregulation of the NADP+ dependent IDHC and ALDH enzymes. Was it not possible? Could they do it?

As noted by the reviewer, in the original version of the manuscript there was no reference about the transcription factors that could be involved in the switch of NADPH source. This was because the up-regulation of Idp2p (IDHC) occurs simultaneously to the onset of the glyoxylate cycle, thus making it impossible to identify the transcription factors that selectively up-regulate each process. However, as the IDP2 gene interacts with the same transcription factors that are activated during the onset of the glyoxylate cycle it is likely that both processes are intertwined at the transcriptional level. This is now also stated in the text.

As for the down-regulation of the PP pathway enzymes, no transcription factor was identified because, even when the involved proteins are clearly down-regulated between 5.1 and 7.4 h, they do not meet the criterion chosen for differential expression. Thus, they were classified as constant for the transcription factor overrepresentation analysis.